# Mapping global urban land for the 21st century with data-driven simulations and Shared Socioeconomic Pathways

Jing Gao [1] & Brian C. O'Neill [2]

Urban land expansion is one of the most visible, irreversible, and rapid types of land cover/land use change in contemporary human history, and is a key driver for many environmental and societal changes across scales. Yet spatial projections of how much and where it may occur are often limited to short-term futures and small geographic areas. Here we produce a first empirically-grounded set of global, spatial urban land projections over the 21st century. We use a data-science approach exploiting 15 diverse datasets, including a newly available 40-year global time series of fine-spatial-resolution remote sensing observations. We find the global total amount of urban land could increase by a factor of 1.8–5.9, and the per capita amount by a factor of 1.1–4.9, across different socioeconomic scenarios over the century. Though the fastest urban land expansion occurs in Africa and Asia, the developed world experiences a similarly large amount of new development.

[1] Department of Geography and Spatial Sciences & Data Science Institute, University of Delaware, Newark, DE 19716, USA. [2] Pardee Center for International Futures & Josef Korbel School of International Studies, University of Denver, Denver, CO 80208, USA. ✉email: jinggao@udel.edu

Urban area is a primary nexus of human and environmental system interactions. Where and how new urban lands are built result from social, demographic, and economic dynamics[1,2], and transform many aspects of the environment across spatial and temporal scales, including freshwater quality and availability (through hydrological cycles)[3], extreme precipitation and coastal flooding (through atmospheric dynamics)[4], biodiversity and habitat loss (through ecological processes)[5], and global warming (through energy use and carbon emissions)[6]. Meanwhile, with global population already 55% urban and becoming more urbanized over time[7,8], cities globally are driving forces of economic value creation and income generation, playing essential roles in many critical social issues[9]. The spatial distribution of urban land also shapes the societal impacts of environmental stresses, such as human exposure to and health consequences of air pollution, heatwaves, and vector-borne diseases[10].

To better understand the future of urbanization and inform new urban development, many have argued for the need of global, long-term, spatial projections of potential urban land expansion[11,12]. As a global trend, urbanization interacts with many large-scale forces like economic globalization and climate change to affect human and earth system dynamics across scales[13]. As one of the most irreversible land cover/land use changes, an urban area, once built, usually remains for the long term without reverting to undeveloped land, and casts lasting effects on its residents and connected environments[14]. Further, spatial patterns in addition to aggregated total amounts determine how urban land patches interact with broader contexts[15].

However, existing spatial urban land projections are usually limited to short-term futures and/or small geographic areas[16,17]. As a result, integrated socio-environmental studies of longer-term trends and larger-scale patterns have often assumed static spatial urban land distribution over time or relied on simple proxies of urban land[12,18], disregarding influential spatial and temporal variations in the urbanization process. Efforts to capture such variations in large-scale spatial urban land models have been impeded by the longstanding lack (until recently) of global, spatial time series of urban land monitoring data. As a consequence, urban land change research about global trends has developed to examine samples of existing cities or city regions[11]. During the past decades, important new knowledge has emerged from this kind of local-scale urban studies and their meta-analyses, but due to the limited scope of their data foundations, the findings remain difficult, if not impossible, to incorporate in large-scale forecast models, creating a gap between qualitative understanding and quantitative models of contemporary urbanization for large-scale, long-term studies.

For example, exiting urban theories derived from studies of individual cities have illustrated that urbanization is a local process that can be motivated by different drivers in different places, and the same drivers in the same geographic area can show distinctively different effects during different maturity stages of urbanization[19]. However, such studies cannot provide information on when and where one type of urbanization process transitions to another for large-scale modeling, and no existing global urban land model captures these well-acknowledged spatial and temporal variations: Existing spatial projections[17,20] have used a single model for all times and all world regions (16 or 17 regions) to project regional total amounts of urban land, without differentiating urbanization maturity stages. The regional totals were then allocated to grid cells using spatial models based on a single-year snapshot map of urban land cover. With no calibration to information on change over time, the models often struggle to identify locations where urban expansion likely occurs and tend to allocate new land development to places with high densities of existing urban land[21]. These modeling efforts also assumed the spatial distributions of key drivers remain static over time, limiting their credible applications to near- to mid-term futures.

A related branch of global models focuses on national totals, leaving out subnational spatial variability. Angel et al.[1] used multiple regression models to investigate how different drivers affect national total amount of urban land, but did not account for changes in urbanization maturity over time nor different urbanization styles across countries. Li et al.[22] used 22-year national data to parameterize for each country a classic sigmoid model, assuming the S-shaped curve would implicitly capture different urbanization stages for the country. However, their results suggest the model may not be responding to drivers in expected ways, as it generated a large amount of urban expansion globally in a scenario intended to represent sweeping sustainability trends across sectors (including sustainable land use). This result confirms the known notion that classic models may need significant structural changes to correctly capture key variations of contemporary urbanization[2,11].

The absence of empirically grounded, large-scale, long-term, spatial urban land projections has obscured our outlook for anticipated global change for a wide range of fields and issues, and hindered the research communities' ability to inform global policy and governance debates concerning urbanization. To fill this gap, we conducted assorted data-science analyses of 15 best available global datasets of urbanization-related socioeconomic and environmental variables at multiple scales, including a newly available global spatial time series of urban land observations[23]. Based on Landsat remote sensing, the data offer the finest spatial resolution (38 m) and the longest time series (40 years: 1975–2014) possible for global urban land observations (defined as built-up land). In contrast, the previous best available global data[24], MODIS land cover type, offers yearly snapshots at 500 m spatial resolution since 2001. With the recent data advances, we quantified spatial and temporal variations of urbanization in ways that can be directly incorporated in the construction of an empirically-grounded urban land change model. Some of the trends discovered update commonly-held beliefs. The advances in modeling allowed us, for the first time, to produce potential long-term futures of global spatial urban land patterns that are empirically calibrated to generalizable trends evident in historical observational data. To account for uncertainties associated with the long-term future, we used the modeling framework in combination with Monte Carlo experiments to develop five scenarios consistent with the Shared Socioeconomic Pathways (SSPs)[25]. This work builds new capacity for understanding global land change with unprecedented temporal, spatial, and scenario dimensions. More importantly, it paves the way for better integrated studies of socio-environmental interactions related to urbanization. Below in the "Results" section we briefly describe the new data-science-based models, and present trends and patterns seen in our projections. For more information on model development and validation, please refer to the "Methods" section and ref. [21].

## Results

**New data-driven urban simulation models**. We developed a pair of models reflecting the observed spatiotemporal patterns at national, subnational regional, and spatial scales. Our national model, Country-Level Urban Buildup Scenario (CLUBS), captures macroscale effects of population change and economic growth on the overall urbanization level of countries. It distinguishes three empirically-identified urban land development styles that occur in countries with different urbanization maturity: rapidly urbanizing, steadily urbanizing, and urbanized. Each

style has its own unique model trained using different drivers and different model parameters determined by historical data. The national model, at the beginning of every decade, uses an empirical classifier to decide for each country which one of the three styles should be used for estimating its decadal total amount of new urban land development, according to the country's urbanization maturity at that time point. As the country's condition changes over time, it may switch styles from decade to decade, and every country's evolutionary path of urbanization is tracked. Our spatial model, Spatially-Explicit, Long-term, Empirical City developmenT (SELECT)[21], is also empirically oriented for long-term projections. It estimates, for 1/8° (roughly 14 km at the Equator) grid cells, the decadal change in the fraction of urban land within each grid, using temporally evolving spatial drivers. SELECT accounts for both subnational regional- and local-scale heterogeneity in the urbanization process through multiple intentionally designed model features, including dividing the world into 375 subnational regions and modeling their spatial patterns separately. For example, the continental United States is modeled as 28 separate regions and China 26 regions (in contrast to existing models' 16 or 17 regions globally). The number of regions (i.e., 375) is not arbitrarily predetermined, but rather through a data-driven delineation reflecting the impact zones of existing cities. The national decadal total amount of new urban land is distributed first among the subnational regions and then grid cells within each region. A unique model is trained for every region using drivers most relevant for explaining the region's observed change in spatial urban land patterns. Models from different regions can use different drivers and different model parameters reflecting their respective past patterns. The structure and validation of SELECT is fully documented in ref. [21]; a summary is provided in "Methods".

Using these models, we developed five urban land expansion scenarios corresponding to SSPs 1–5, named as sustainability, middle of the road, regional rivalry, inequality, and fossil-fueled development. The scenarios span a wide range of uncertainties in drivers of urban land development. The urban land projections differ across scenarios due to both differences in drivers and varying model parameter values selected from Monte Carlo experiments to be consistent with our interpretation of urbanization styles implied by the SSP narratives. The resulting projections therefore cover a broad spectrum of plausible urban futures.

**Global trends**. The projections show the amount of urban land on Earth by 2100 could range from about 1.1 million to 3.6 million $km^2$ across the five scenarios (roughly 1.8–5.9 times the global total urban area of about 0.6 million $km^2$ in 2000). Under the middle of the road scenario, new urban land development amounts to more than 1.6 million $km^2$ globally, an area 4.5 times the size of Germany. Global per capita urban land more than doubles from 100 $m^2$ in 2000 to 246 $m^2$ in 2100.

The global total amount of urban land strongly depends on societal trends in years to come (Fig. 1). The lowest amount of global urban land was projected for the sustainability scenario, due to low population growth, the rise of less resource-intensive lifestyles, and international collaborations emphasizing global environmental and human well-being. The highest scenario is fossil-fueled development, due to high population growth, accelerated globalization driven by material-intensive economies, and low concern for global environmental impacts—all stimulating sprawl-like development. These factors influence the urban land projections both as quantitative drivers in the model, and by qualitatively determining what trajectory (high, medium, or low) a scenario follows within the uncertainty

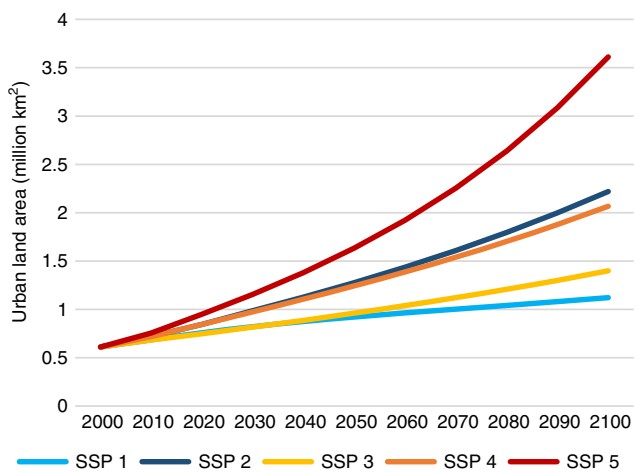

**Fig. 1 Global total amount of urban land under different scenarios over the 21st century.** The scenarios correspond to the five Shared Socioeconomic Pathways (SSPs 1–5): sustainability, middle of the road, regional rivalry, inequality, and fossil-fueled development.

range generated by Monte Carlo simulations (more information in "Methods").

Results also indicate that similar amounts of urban land may be produced by scenarios with different driving factors. For example, sustainability (SSP 1) and regional rivalry (SSP 3) both have relatively slow urban land expansion. In sustainability, low population growth and preference for environmentally friendly lifestyles reduce the demand for urban expansion, and improved green technologies can make any new development more compact. In contrast, in regional rivalry, economic and technological developments are impeded, leaving countries little means to develop new urban land, even though the prominent consumption style in the scenario is material-intensive. Similarly, middle of the road (SSP 2) and inequality (SSP 4) also show similar amounts of urban expansion. In the middle of the road scenario, urbanization drivers follow historical patterns, without substantial deviation from central trajectories within their respective uncertainty ranges. In inequality, low-income countries follow a slow urbanization trajectory due to poor domestic economic development, low internal mobility, and lack of international investment, while more economically developed countries follow a medium trajectory at the national level, an aggregate of within-country heterogeneity across faster and slower developing urban areas.

**Regional variations**. At the regional level, we find that all world regions, not just developing regions, can experience more than 4.5-fold urban land expansion in the high urbanization scenario (SSP 5) by 2100 (Supplementary Table 1). Under the middle of the road scenario, urban land in Europe (excluding Russia, which is a separate region in this analysis due to its large land area) is projected to expand by more than 275 thousand $km^2$, more than outpacing Africa's increase of about 193 thousand $km^2$. These projections result from an often overlooked pattern in observed land change trends: economically developed regions have not stopped building new urban land. For example, according to our own analysis of time series observational data[23,26], during the decade of 2000–2010, more than 15 thousand $km^2$ of new urban land was built in Europe (excluding Russia), and 17 thousand $km^2$ in Africa. These similar amounts of urban land development occurred despite the fact that in 2000, Europe already had more than 144 thousand $km^2$ urban land (with about 0.6 billion people), while Africa had only about 46 thousand $km^2$ urban land (with about 0.7 billion people). Many European countries have

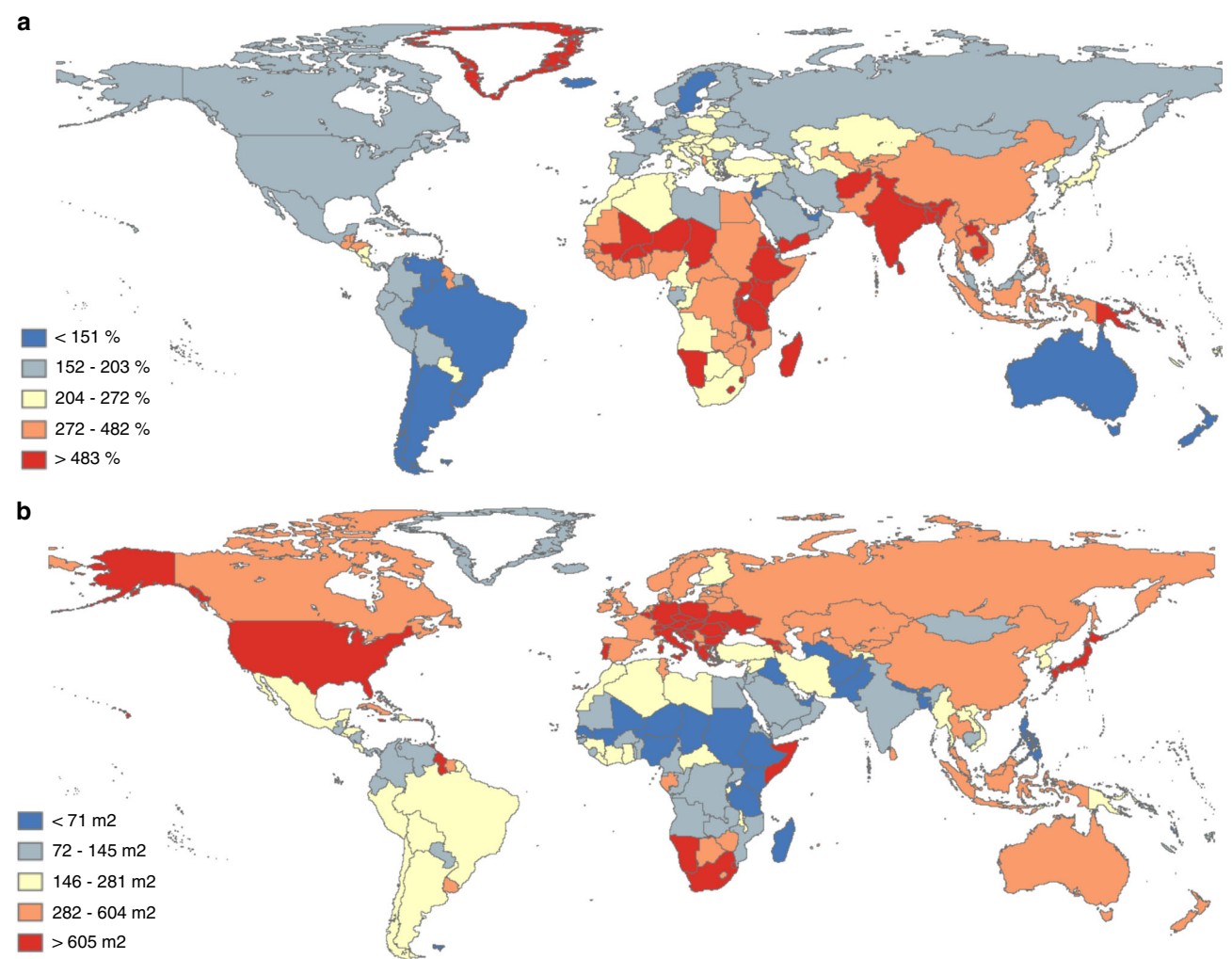

**Fig. 2 National urban land expansion in the middle of the road scenario. a** Global quintile map of urban land expansion rate (%) 2000–2100. **b** Global quintile map of per capita urban land area (m$^2$) in 2100 (this variable in 2000 is shown in Supplementary Fig. 1). (Source data are provided as a Source Data file).

been consistently expanding urban land areas after their population growth has stopped. Although the change rate in percentage terms is much lower than what is seen in Africa and Asia, the absolute amount of new urban land development is substantial, reflecting the much larger amount of existing urban land in the developed world. This observed trend led to the result that, depending on societal trends and choices, a large amount of new urban land might be built in the developed world during the 21st century (Figs. 2 and 3, Supplementary Fig. 2). Two dynamics observed in the recent past may help explain such urban expansion patterns. First, even in the regions and countries whose total population size decreased, their urban population size continued to grow[7]. Over the 21st century, we expect to see more population concentrate in urban areas globally[7,8], and new urban land will be built to support this change. Second, though functional types of urban land are not explicitly distinguished in this work, a sizable fraction of urban land development globally is non-residential, e.g., built-up land used for industrial, commercial, and institutional purposes. These developments do not necessarily scale with population size, and might be driven by shifts in economy, governance, livelihood, culture, and lifestyle as urbanization matures[27–30]. Although these abstract factors are not explicitly modeled, their effects may be represented in empirical models through implicit relations

with measurable variables (e.g., GDP, urban population share). Such relations, if present during historical times, are likely to continue affecting urban land development after population size stabilizes.

In contrast, developing countries in Africa and South Asia continue to show the lowest per capita levels of urban land over the century compared to the rest of the world (Fig. 2b), despite having the highest projected urban land expansion rates. These findings modify the common belief that the developing world is the primary realm of urban expansion. We find that both developed and developing countries will play substantial roles in shaping the planet's urban future.

**National styles**. At the national level, historical data showed three distinctive urban land expansion styles driven by different socioeconomic dynamics, and global countries evolve through the three styles directionally from rapidly-urbanizing to steadily-urbanizing to urbanized (Fig. 4, Supplementary Table 2). Present-day rapidly-urbanizing countries are primarily low-income developing countries with the most vulnerability to social and environmental stress, steadily-urbanizing countries are already moderately urbanized developing countries transitioning into more stabilized urbanization phases with lower yet steady urban

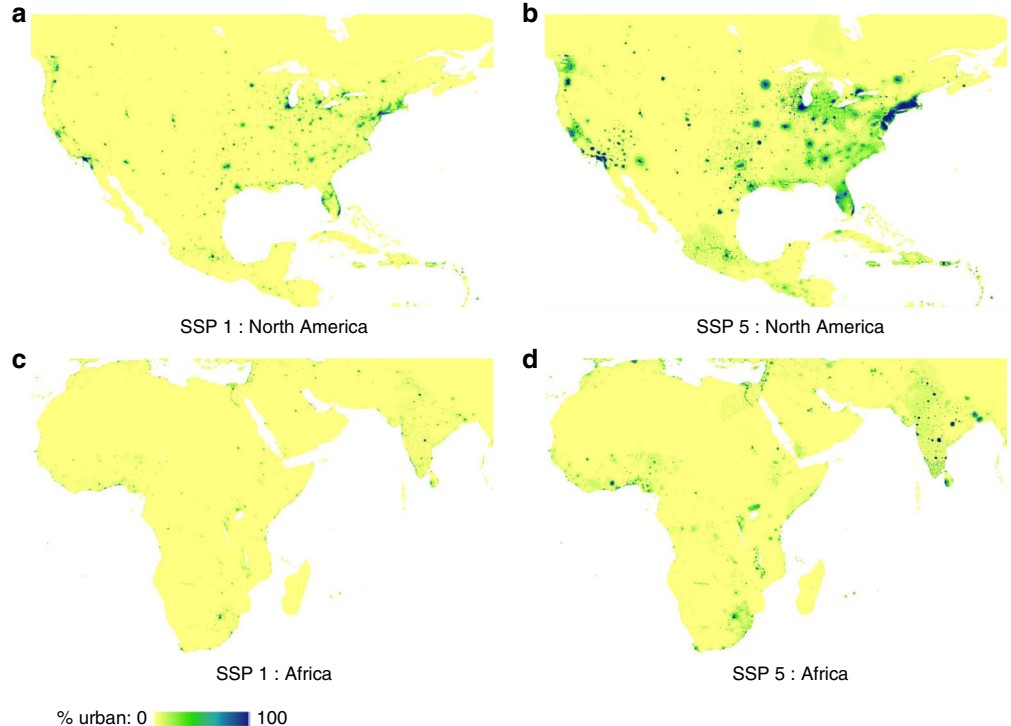

**Fig. 3 2100 spatial urban land maps.** This figure compares the sustainability scenario (SSP 1) and the fossil-fueled development scenario (SSP 5) for the most developed and the fastest developing continents (North America and Africa, respectively). **a** North America under SSP 1. **b** North America under SSP 5. **c** Africa under SSP 1. **d** Africa under SSP 5.

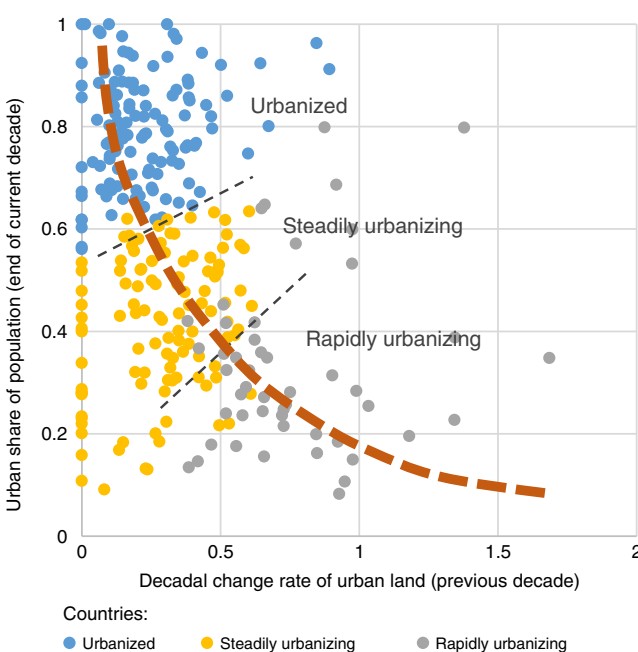

**Fig. 4 Three styles/maturity stages of urban land expansion.** This scatter plot of decadal observations of all countries concurrently shows data from three different decades (1980–2010). The two axis variables are shown for visual clarity. The actual classification step in the modeling framework uses more variables (more information in "Methods"). (Source data are provided as a Source Data file).

change rates (at present both India and China are in this category), and urbanized countries comprise economically developed countries and some highly urbanized developing countries (e.g., Brazil).

Table 1 shows the empirically-determined coefficients of relevant drivers in the three models for different urbanization styles, which include different aspects of urban or total land, urban or total population, and GDP. The results indicate two patterns: first, the three urbanization styles are driven by different sets of factors, and second, the same driver casts different effects on different urbanization styles. Urban land expansion in rapidly-urbanizing countries is driven by total population change (less so by urban population change) indicating that their urban land development has strong ties with basic infrastructure needs that scale with population size. These countries are the least urbanized in the world, and changes appear faster (i.e., higher change rates) when existing urbanization levels are lower (i.e., the base values are smaller). Urban land expansion in steadily-urbanizing countries carries significant inertia from the previous decade. As a transition style, it responds to both factors affecting rapidly-urbanizing countries and those affecting urbanized countries. Most significantly, steady urbanization of land is strongly coupled with urban population increase. Finally, urban land expansion in urbanized countries shows the flattest change curve and responds faintly to urbanization of population and GDP change. Moreover, the GDP effect carries a negative sign, i.e., countries with the fastest GDP growth tend to have lower urban land expansion rates, suggesting that the fastest growing economies are less reliant on new urban land development. This may also relate to effects of globalization where high-income countries may have their needs for urban land intensive industries (e.g., production factories) met by such industries located in lower-income countries[31].

By 2100, most countries globally become urbanized under most socioeconomic scenarios, while the timing of how soon currently developing countries transition to more stabilized urbanization trajectories depends on societal trends (domestic and international) reflected in the scenarios (Supplementary Tables 3 and 4). For example, in scenarios with slow population growth, most

**Table 1 Drivers of urban land expansion and their effects on different styles of urbanization, shown by standardized coefficients of linear models trained for the three styles.**

|  | Urbanized | Steadily urbanizing | Rapidly urbanizing |
|---|---|---|---|
| Change rate urban land (previous decade)[a] | 0.24 | 0.35 | 0.17 |
| Urban share of population (end of decade)[a] | −0.15 | −0.2 | −0.48 |
| Change rate urban population share (current decade)[a] | 0.09 | 0.25 | 0.09 |
| Change rate GDP (current decade) | −0.09 | −0.1 | |
| Change rate population size (previous decade) | | 0.12 | 0.13 |
| Land area | | 0.08 | −0.14 |

[a]These variables are used by a k-means clustering algorithm trained on historical data, to classify each country at the beginning of a decade into one of the three urban expansion styles.

countries in the rapidly-urbanizing category at the beginning of the century remain in that category throughout the century. In contrast, in other scenarios no countries remain in the rapidly-urbanizing category by mid-century (Supplementary Table 4). Meanwhile, for urbanized countries, although much flatter change curves apply compared with urbanizing countries (Fig. 4), effects of different socioeconomic trends can accumulate over time and lead to substantial differences in spatial urban land patterns across scenarios (Fig. 3, Supplementary Fig. 2). In 2100, both developed and developing countries show roughly three times as much urban land in the fossil-fueled development scenario as in sustainability (Supplementary Tables 1 and 5). That is, moderate amounts of urban land expansion are possible for the 21st century, but will require intentional planning and societal choices in both developed and developing worlds.

**Spatial patterns**. At subnational, local levels, our model is the first to be empirically grounded by observed historical spatial changes, and its projections show distinctively different spatial patterns from results of existing global urban land change models. Most prominently our projections allocate new urban land development to places with similar characteristics to where urban expansion happened in the observed past, while existing global urban land projections tend to allocate new urban development to places with high existing urban land densities[21].

Our projections captured subnational regional variations in spatial patterns of new urban land development. For example, within the Unitd States, new urban development is projected to expand broadly across space along the southeast coast, intensify urban land density within clusters of existing cities in the northeast, and emerge as new smaller settlement sites in the southwest (Fig. 3, see the high urbanization scenario). These projections reflect observed historical patterns of how urban land tends to change in these subnational areas, and add novel spatial details to global urban land modeling.

Our model's ability to project with subnational, local-scale variations also allowed it to make educated guesses about where new large cities might emerge in the future. Existing literature has established that many existing small settlement sites might become large or mega cities over the 21st century, due to the observed pattern that more urban expansion occurs around medium- to small-sized settlement sites globally[7]. Our projections identified potential locations of such booming towns. Currently small settlement sites that have exhibited fast urban expansion are projected to continue growing rapidly, and often outpace existing larger but slowly growing cities, to become sizable future urban centers. See, for example, the emergence of new sizeable cities in India under the high urbanization scenario (Fig. 3). Although global long-term projections should be taken with a grain of salt in highly local applications, our projections can be considered a potential indicator for possible future development hotspots.

The spatial patterns of our urban land projections vary substantially across scenarios (Fig. 3, Supplementary Fig. 2). The differences are affected by both national total amounts of urban land and spatiotemporal trends in drivers of urbanization. The model was able to capture the spatially-explicit divergence among scenarios because it updates key spatial drivers every time step (i.e., one decade) and these drivers reflect how urban land patterns have evolved locally at previous time points. This allowed spatial patterns under different scenarios to evolve through different pathways, in contrast to some existing projections that derive different scenarios by scaling a single spatial pattern. For example, in Fig. 5, the overall amount of urban land in the displayed region is similar for SSP 2 in 2100 and SSP 5 in 2060, but the two maps exhibit different spatial patterns: Under the middle of the road scenario, urban expansion around New York City is confined and other smaller cities in the region experience more growth, while under the fossil-fueled development scenario, all city areas show expansive sprawl—consistent with the narratives of the two scenarios.

Though there are no data for validating long-term spatial projections, we tested our modeling framework as thoroughly as data permit. Our spatial model (SELECT) explained high fractions of variations in its response variable for all 375 subnational regions across the world with low estimation residuals in short-term applications, and showed advantages when compared with an example existing urban land change model for making mid-term projections[21]. To gain confidence in the model's long-term reliability, we also tested its statistical robustness by examining the model's reaction to simulated noise, spatial generalizability by swapping models trained for different subnational regions, and temporal generalizability by leaving one decade of historical data out of model training for independent performance evaluation. The model scored satisfactorily in all tests we ran, and a complete report of these validation results has been published[21]. In addition, we examined the plausibility of our future projections by comparing projected trends and patterns with urban land change theories already established by existing literature. For example, cities across the world are generally becoming more expansive[2], i.e., they grow faster in land area than population size. In our medium to high development scenarios, most world areas continue to show higher change rates for urban land than population, leading to increases in per capita amounts of urban land, while in low development scenarios parts of the world reverse the historical trend to show more compact urban land use (Supplementary Tables 6, 1, and 5). The world regions with reversed trends are different under different scenarios but coherent with their respective narratives. Though fidelity to known aggregate patterns like this should be a basic requirement for projection models, it often cannot be assumed, especially when the time horizon to be projected is much longer than the one in available training data. Altogether, the results from model

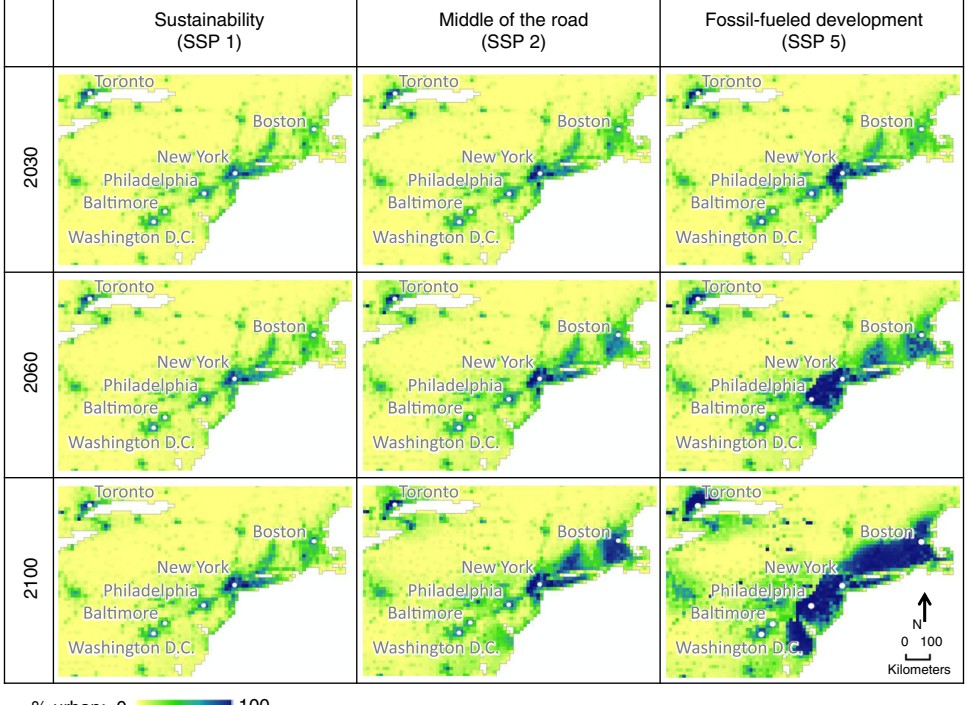

**Fig. 5 Temporal pathways of urban expansion in northeastern United States under different scenarios.** These maps show how different spatial patterns evolve under various scenarios through different pathways. Spatial patterns can differ, even when the overall amount of urban land in the displayed region is similar, e.g., SSP 2 in 2100 and SSP 5 in 2060.

validations and plausibility tests give confidence in the quality of our projections, and also illustrate the potential of creative data-science applications for studying complex social processes.

## Discussion
Our results demonstrate that a wide range of outcomes for urban land extent are plausible over the coming century, both in terms of aggregate amounts and spatial patterns. Given the complexities of urban land change, the global scope, the long time horizon, and data limitations (even with recent improvements to data availability), any approach to such modeling efforts will have both strengths and limitations.

To maximally take advantage of all information available, we integrate existing theories and new data. On balance, the method is data-driven, because the newly available time series of global spatial urban land we use here offer perspectives that were not possible before, and we therefore allowed observational evidence from the data analyses to extend and update existing theories. However, we view the perspectives offered by existing theories and new data as a gradient of concepts rather than a black-and-white distinction. Theory-based stylized-fact models often need to be empirically calibrated and adjusted to be able to generate realistic patterns[32], and classic machine learning theories such as the bias-variance decomposition have long recognized the necessity to incorporate prior knowledge in the structural design of data-driven models when modeling complex phenomena[33,34], which long-term global urbanization certainly is. Below we highlight two examples of how existing theories and new data are combined in our methods, while the integrative principal was applied throughout our model design.

Example 1: At the national level, we took lessons from the existing literature that relationships between urban land change and its drivers change over time and usually do not scale linearly. While existing theories have not provided insights on how many

urbanization styles are present among countries across the world and how they transition from one to another, our data-driven analysis found three unique urbanization styles. Examining the characteristics of countries falling into each style/cluster, it was clear that the distinction among clusters is linked to urbanization maturity. Utilizing these findings, CLUBS can organically model how global countries change urbanization styles as their urbanization matures, without needing arbitrary input from the analyst. The data-based new insights enabled improvements to conventional methods that often fit a simple linear model for less than twenty world regions to capture the same process.

Example 2: When modeling subnational spatial patterns of urban land, to account for the theoretically-grounded understanding that primary drivers of spatial urban land change are different in different places, we used data to help divide the world into 375 distinct subnational regions for which independent spatial models were developed, while making the set of driving variables entering the model as inclusive as possible according to existing literature. We then used a highly flexible nonparametric statistical-learning method to determine the relationship between urban land development and each driver in each subnational region, so that different regions are modeled by different subsets of all input variables using different relationships. The automated model fitting process also frees the analyst from having to manually and consistently parameterize close to four hundred different models. In contrast, conventional methods treat the world as less than twenty regions, and do not capture subnational, regional variations in urbanization.

A challenge for all long-term modeling of socioeconomic variables is the necessity to assume some temporal nonstationarity while the underlying process is bound to change over time. We addressed this challenge by incorporating a transition mechanism in the national-level model allowing countries to change from one national urbanization style to another as their

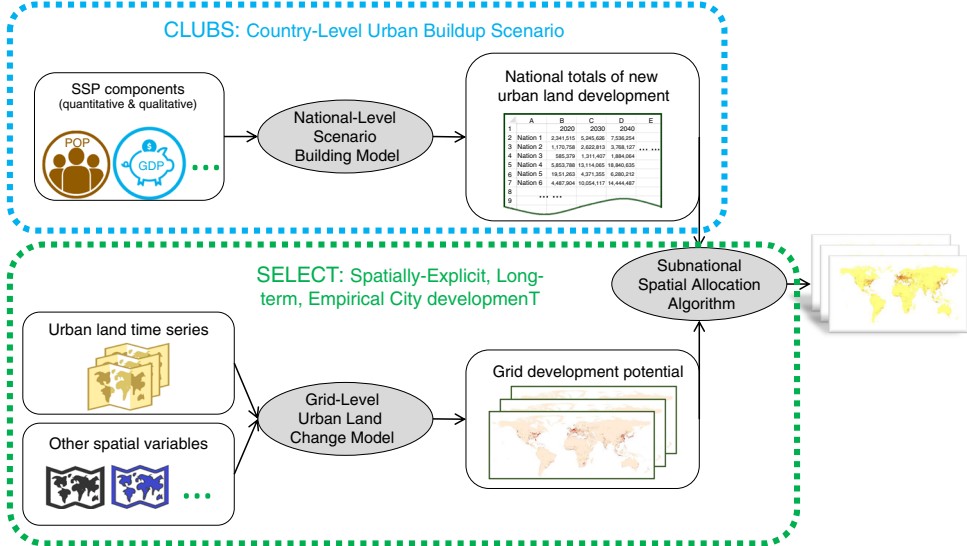

**Fig. 6 Modeling framework.** This framework consists of two new data-driven urban simulation models: The Country-Level Urban Buildup Scenario (CLUBS) model estimates national total amounts of new urban land development under various socioeconomic scenarios. The Spatially-Explicit, Long-term, Empirical City developmenT (SELECT) model allocates the national totals, first to subnational regions, and then to 1/8° grid cells according to their estimated development potentials. Both models evolve over time decade by decade. (Gray ovals—model parts; white squares—input data and intermediate output).

urbanization processes evolve. In addition, we used different sets of parameters according to different SSP-based scenarios for producing alternative urban land development patterns to capture a wide range of plausible future uncertainties. Nonetheless, the spatiotemporal dynamics that induce changes to spatial patterns for each subnational region stay the same over time. While this is a limitation of our method, we still consider it an improvement over existing models: In comparison to the commonly-used assumption that new urban development is primarily attracted to areas with high amount of existing urban land (which may have not changed for decades), our model allocates new development primarily to areas sharing similar characteristics to places of recent, rapid urban expansion, which is a substantially less constraining approach.

Our method is also well suited to developing alternative urban land change scenarios, given its responsiveness to different possible trends in drivers as well as its ability to cover varying parametric uncertainties. Currently, the projected scenarios do not incorporate the impacts of potential future environmental change (e.g., sea level rise, climate change) on urbanization. Because studying such impacts is one of the scenarios' planned uses, the impacts are excluded so that the SSP-based scenarios can be used as references. This is consistent with SSP practice in other domains. However, future work could improve the plausibility of projected urban land patterns by accounting for potential influence from environmental change.

Finally, as we anticipate continued urban land expansion in the coming decades, understanding long-term impacts of alternative urban land patterns can inform policy and planning strategies for achieving future sustainable development objectives. Our results, showing a wide range of uncertainties in future urbanization, underscore the need for improved understanding of interactions between urbanization and other long-term global changes in society, economies, and the environment. Our projected scenarios enable integrated modeling and studies of these interactions, and can help researchers and policy analysts identify potential pathways towards desirable urban outcomes in a globally changing environment. After all, land use is not destiny. The planet's urban future is being shaped by development happening today, and the time to act for better urban policy, planning, design, and engineering is now.

## Methods

**Modeling framework.** With large-scale investigations of human–environment interactions in mind, we designed a two-tier modeling framework (Fig. 6) usable in conjunction with global environmental and climate modeling results. It consists of the CLUBS model, and the SELECT model. The modeling framework uses globally-consistent, multi-scale, spatially-explicit data, reflects the local, spatially variant nature of the urbanization process while maintaining global coverage, functions well over long time horizons, and offers means to generate scenarios considering alternative development trajectories.

We projected the fraction of potential future urban land for 1/8° grid cells covering global land at 10-year intervals throughout the 21st century. The 1/8° resolution was chosen as a balance between the very different commonly-used spatial resolutions of global change models and local urban studies: Global change modeling (e.g., climate modeling) usually functions at much coarser spatial resolutions (e.g., 0.5–2°) and longer time horizons (e.g., throughout the 21st century) than common urban land change models (e.g., 30–500 m, up to 30 years). The choice is also a balance between the different spatial resolutions of available input data (Table 2). To capture the often incremental urban land expansion with local-scale precision, we used the fraction of urban land within each 1/8° grid cell as the response variable, rather than the more commonly used binary variable (urban vs. non-urban) or the probability of conversion (of entire grids) in conventional urban land change studies. The 1/8° urban land fractions for historical times (1980–2010) were derived from a global 40-year (1975–2014) time series of 38-m Landsat remote sensing based urban land observations[23]. The fine-spatial resolution of the base data gave the fraction estimates the most precision currently possible. Changing from a categorical response variable to a numerical one also alters the set of analytical tools available for model development, another way in which this modeling framework differs from conventional methods.

In this work, urban land is defined as the Earth's surface that is covered primarily by manmade materials, such as cement, asphalt, steel, glass, etc. This land cover type is referred to by different communities using different terms (with nuances), such as built-up land, impervious surface, and developed land. Remote sensing is an ideal source of global observations for this land cover, but has shown less success identifying settlements made of natural materials similar to their surrounding environments. As an observational technique, it also tends to underestimate development of really low density, e.g., exurbanization. Nonetheless, our input data, by having a much finer spatial resolution (38 m) than previous commonly-used large-scale urban land cover data[35] (MODIS at 500 m), are less affected by these issues although not immune. A few other remote sensing based datasets reporting urban land have also become available in recent years, for example, the European Space Agency (ESA) Climate Change Initiative's (CCI) Land Cover project[36] maps global land cover at 300 m for 1992–2015, and the German Aerospace Center's (DLR) Global Urban Footprint dataset[37] offers a 12 m snapshot around 2012. However, none provides as long of a time series at as

**Table 2 Key model variables and their data sources.**

| Key variables | Training data source | Projection data source |
|---|---|---|
| Spatial and national urban land time series | Global Human Settlement Layer[23] (38 m, decadal 1980–2010[b]) | PROJECTIONS (1/8°, decadal 2000–2100) |
| National population size[a] | U.N. World Population Prospect[38] (national total, decadal 1980–2010) | SSP National Population Count Projections[39] (national total, decadal 2000–2100) |
| National urban population share[a] | U.N. World Urbanization Prospect[7] (national total, decadal 1980–2010) | SSP National Urban Population Projections[8] (national total, decadal 2000–2100) |
| National GDP[a] | OECD National GDPs[40] (national total, decadal 1980–2010) | SSP National GDP Projections[40] (national total, decadal 2000–2100) |
| National scenario trajectory setting (Monte Carlo experiment)[a] | | SSP Narratives[25] (qualitative descriptions of trends 2000–2100) |
| Spatiotemporal texture of urban land change | Global Human Settlement Layer[23] (38 m, decadal 1980–2010) | Updating with PROJECTIONS (1/8°, decadal 2000–2100) |
| Spatial population count time series | Gridded Population of the World (v.4 and 3)[26] (1 km, decadal 2000–2010) | SSP Spatial Population Projections[41] (1/8°, decadal 2000–2100) |
| Topographic contexts (elevation, slope) | Global Multi-Resolution Terrain Elevation[42] (1 km, snapshot) | Static over time |
| Distance to waterbodies | World Waterbodies[43] (1 km, snapshot) | Static over time |
| Distance to existing cities (with >300k ppl) | U.N. World Urbanization Prospect[7] (latitude–longitude coordinates, snapshot) | Static over time |
| Developable land mask | SSP Spatial Population Projections[41] (1/8°, snapshot), Global Rural-Urban Mapping Project[43] (1 km, snapshot) | Static over time |

[a]These rows are used by CLUBS. The rows at the bottom half of the table are used by SELECT.
[b]GHSL has four time points—1975, 1990, 2000, 2014—we used temporal linear interpolation to generate maps for 1980 and 2010, so that the time steps are regular and the time points align with other datasets.

fine of a spatial resolution in combination. We therefore consider our input data the best available for globally-consistent spatially-explicit time-series observations of change in urban land.

**CLUBS.** At the national level, CLUBS generates scenarios estimating for each country how much total new urban land development would happen per decade of the 21st century, driven by inputs from existing SSP components, including quantitative projections of population size, urban population fraction, and GDP, as well as qualitative narratives describing alternative future societal trends (Table 2). CLUBS's model structure was designed to reflect robust relationships identified by mining historical data on urban land expansion, demographic change, and economic growth.

We first ran multiple feature selection methods (lasso regression, regression tree, and correlation matrix) on 99 different metrics of (change in) urban land, GDP, population size, and urban population share, and 60 variant subsets of these metrics. Example metrics for measuring urban land include national total amount of urban land, national total amount of change, national change rate, per capita urban land, per capita change, per capita change rate, at 10-, 20-, 30-year intervals, calculated for various periods within three decades (1980–2010) of historical data. We found that the most robust statistical relationships across space and time occur among national change rates of urban land, demographic change, and economic growth (i.e., how fast each variable changes at national level) measured over 10-year intervals, rather than the commonly-used per capita urban land and per capita GDP.

We also ran multiple clustering analysis methods (hierarchical clustering, k-means clustering, and decision tree) on the different combinations of the different metrics mentioned above. The analysis uncovered three urban land expansion styles (rapidly urbanizing, steadily urbanizing, and urbanized) as three unique clusters, and the country-decade combinations (e.g., US 1980–1990, Ethiopia 1990–2000) can be classified into these clusters using three descriptive variables (Table 1). We trained a k-means classifier using these variables from historical data (1980–2010). The analysis results show that common geographic regions do not separate the different urbanization styles well, and over time countries evolve through the three styles from rapidly-urbanizing to urbanized (Supplementary Table 2). As some existing literature[19] has alluded to, we found that the same unit change in population or GDP show different effects on the same country at different times, and on different countries within the same geographic region (e.g., Singapore and Indonesia show distinctively different urban land expansion trajectories). When making future projections, each country is classified, at the beginning of every decade, into one of the three urban land expansion styles. As the country develops over time, it may receive different classifications for different decades and under different socioeconomic scenarios (Supplementary Tables 3 and 4) in response to how the country's urbanization maturity evolves.

For each of the three urban land expansion styles, we trained a unique model using data on country-decade combinations belonging to that urban expansion

**Table 3 Urbanization scenarios corresponding to SSPs 1–5.**

| | | SSP 1 | SSP 2 | SSP 3 | SSP 4 | SSP 5 |
|---|---|---|---|---|---|---|
| 1 | Urbanized | Low | Medium | Low | Medium | High |
| 2 | Steadily urbanizing | Low | Medium | Low | Medium | High |
| 3 | Rapidly urbanizing | Medium | Medium | Low | Low | High |

Nations of different urbanization styles under different scenarios are likely to experience different urban expansion rate trajectories (high, medium, or low) within their respective uncertainty ranges generated by Monte Carlo simulations. SSPs 1–5: sustainability, middle of the road, regional rivalry, inequality, fossil-fueled development.

style during historical times (1980–2010). If a country experienced urban expansion style A during 1980–1990 and urban expansion style B during 1990–2000, its data over 1980–1990 is used to train style A model and its data over 1990–2000 is used for style B model. For each urban expansion style, we re-ran feature selection (lasso regression, regression tree, and correlation matrix) and identified the strongest explanatory variables for estimating national decadal urban land expansion rate under that urban expansion style. We tested diverse variants of general linear models and regression trees in search of a modeling method, and chose the simple linear model in the end (Table 1), for its robustness, interpretability, and ability to generate alternative scenarios in a transparent way. Overall, the national model exhibits an $R^2$ of 0.503 for estimating decadal national urban land change rates (i.e., the model's response variable), and if we translate that to end-of-the-decade national total urban land areas, the $R^2$ would be 0.998. The high value of the latter $R^2$ is partially due to the low magnitude of decadal changes in national total urban land areas relative to existing amounts, which is also evidenced by the fact that the $R^2$ of a dummy baseline model that assumes constant national change rates is 0.983 on the same variable.

We developed SSP-consistent urban land expansion scenarios by combining qualitative interpretation of existing SSP narratives, and quantitative simulations using Monte Carlo experiments. Existing SSP narratives describe alternative trends of future population change, economic development, political governance, and technological progress, but not urbanization explicitly. Considering the SSP narratives as global socioeconomic contexts, we determine and assign for each of the three urban land expansion styles whether they are likely to experience high, medium, or low urban land expansion rates within their respective uncertainty ranges in different scenarios (Table 3). For example, under SSP 4, inequality is prevalent within and across countries. Rapidly urbanizing countries (which are

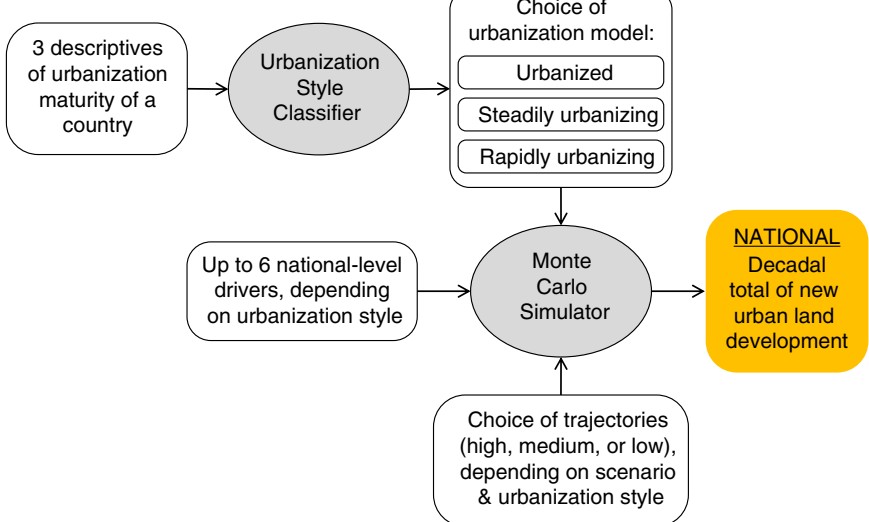

**Fig. 7 Decadal update routine of the Country-Level Urban Buildup Scenario (CLUBS) model.** This flowchart shows how CLUBS estimates the national total amount of new urban land development for one country over one decade. The process repeats for all countries, and iterates over time decade by decade. (Gray ovals—model parts; white squares—input data and intermediate output; orange square—final output).

usually low-income countries) are expected to experience poor domestic economic development, low internal mobility, and low international investment; hence, their urbanization progress is expected to follow a slow trajectory. Meanwhile, steadily urbanizing and urbanized countries are expected to follow a medium trajectory under the same scenario at the national level, averaging domestic heterogeneity between faster and slower developing urban areas.

One Monte Carlo experiment was conducted for each country per decade (Fig. 7). After a country-decade combination is classified into an urban land expansion style, we generate 1000 model variants of the simple liner model for that urban expansion style. Model variants were created by randomly drawing alternative coefficients from normal distributions centered around the estimated coefficients of the simple linear models, and with their corresponding standard errors as standard deviations. These model variants each make an estimate, and together provide an uncertainty range of the country's change rate in total urban land area for that decade. Depending on the country's urbanization style and scenario (Table 3), a high, medium, or low value derived from the Monte Carlo estimates is used as the final estimate. A high value is the mean of the top 80–60% Monte Carlo estimates, a medium value is the mean of the middle 60–40% estimates, and a low value is the mean of the bottom 40–20% estimates.

After every country's Monte Carlo experiment is completed for a decade, CLUBS updates relevant variables, re-evaluates each country's urban expansion style for the next decade, and runs another round of Monte Carlo experiments for the new decade. This process repeats iteratively until time reaches the end of the 21st century.

**SELECT**. SELECT was also developed using a data-science approach and oriented for long-term, global studies of human–environment interactions. Specifics of SELECT, its validations, and a comparison with an existing global spatial urban land model, URBANMOD[17], have been published as a separate open-access paper[21]. Please refer to that for a full report. Below we provide only a brief summary of its essential elements. SELECT is a statistical-learning-based model that consists of the following two model components (Fig. 6).

First, the spatial urban land change model estimates development potentials for 1/8° grid cells covering global land. The model was trained using spatially-explicit time series of urban land change history (derived from fine-spatial-resolution remote sensing observations), and best available global data on spatial population time series and environmental variables (Table 2). The model captures both the globally-averaged general trend of urbanization and locally-unique dynamics affecting spatial patterns of new urban land development. To detect the latter, we divided the world's land into 375 subnational regions according to the locations and densities of existing cities with population sizes greater than 300,000, and modeled each region separately using a non-parametric statistical-learning technique, generalized additive model (GAM). GAM allows the relationships between the response variable and its drivers to be fully determined through model training using observational data, and can "mute" certain drivers for a region if the drivers are proven not useful for explaining spatial urban land changes within that region. As a result, different regions are automatically modeled using different sets of drivers and different relationships according to historical patterns, without

requiring the analyst to manually parameterize hundreds of models for different regions across the world. We designed this model with long-term projections in mind; for example, the way the 375 subnational regions were divided guarantees that each region contains at least one existing sizeable city and a complete spatial gradient of rural–urban transitions, so that when historically undeveloped landscapes become urbanized in the future, the model implicitly uses rural–urban transitions seen during model training at spatially nearby, more urbanized locations as analogies for temporal transitions that the undeveloped landscapes have never experienced, avoiding unconfined extrapolation. The global, long-term scope of this work also limits the number of variables available as drivers for the spatial model. For example, road network is often considered a useful predictor by conventional urban land change models; however, no century-long global projections exist. To account for the effects of these variables and others that are difficult to quantify consistently across the world (e.g., land tenure), we used 108 focal metrics describing the geometric patterns of urban land and how they change over time across a range of local-scale neighborhoods surrounding each 1/8° land grid (more information in ref. [21]) as proxies. The use of these proxies as model drivers is based on the assumption that the present patterns of urban land and how they have changed reflect the collective effects of all driving factors. The proxy drivers are updated at every time step and help capture the temporal evolution of spatial urban land patterns (Fig. 5).

Second, the subnational spatial allocation algorithm distributes exogenously estimated national decadal total amounts of new urban land development (e.g., the estimates made by CLUBS), first to subnational regions based on the expansiveness of each region's urban land use history and population change (which is particularly affected by subnational migration), and then to grid cells within each region proportionally to the development potentials estimated by the spatial urban land change model.

Compared with existing global spatial urban land models, SELECT excels at identifying locations for high future development potentials as places with similar characteristics to observed development hotspots, which may not be places that have a lot of existing urban land at present[21]. This advantage resulted from the fact that SELECT's response variable is a change map rather than a snapshot of urban land, and the model's key drivers are temporally evolving and updated at every time step, while existing global spatial urban land projections[17,20] used static snapshots as both model response and drivers. By doing so, they implicitly assumed that places with existing urban land are attractive for new development, which made the models exhibit behaviors similar to gravity models and are implicitly relying on spatial interactions for generating change in spatial patterns.

In sum, the CLUBS–SELECT modeling framework is unique and improves existing large-scale, long-term spatial urban land modeling efforts through its extensive data-science foundation, strong focus on modeling change, explicit capturing of multi-scale spatiotemporal variations, smooth integration of qualitative and quantitative information, and thorough model evaluation.

**Reporting summary**. Further information on research design is available in the Nature Research Reporting Summary linked to this article.

## Data availability

Datasets produced by this work (i.e., national total amounts of urban land [https://doi.org/10.7910/DVN/85PJ1D], and time series maps of urban land [https://doi.org/10.7910/DVN/ZHMI1L], under urbanization scenarios consistent with the SSPs) are publicly downloadable at dataverse.harvard.edu/dataverse/geospatial_human_dimensions_data. The raw data for all figures displaying contents other than these projections are provided in the Source Data file.

## Code availability

Codes and materials used to produce this work are available upon request.

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

## Acknowledgements

This work was supported by the National Science Foundation [grant numbers 1416860 and 1243095] and the Department of Energy [grant number DE-SC0016438].

## Author contributions

J.G. led model design and development, data analysis and production, result interpretation and manuscript drafting, and contributed to the conception of the project. B.O. led the conception of the project, and contributed to model design, result interpretation, and manuscript drafting.

## Competing interests

The authors declare no competing interests.
