## [Peer Review File · Nature Communications]

Reviewers' comments:

Reviewer #1 (Remarks to the Author):

[General]

The manuscript "Mapping Global Urban Land Expansion for 21st Century Using Data Science Based Simulations" covers a highly interesting and relevant topic and aims at consistent urban land cover scenarios for the 21st century considering different urbanization pathways and states. It is generally well written, yet the provided materials are insufficient to assess if it is technically sound and I am much more sceptic about the results than the authors.

[Major]

- Relatively little detail is given about the used model and just a reference and various input datasets reported are provided. To me it is relatively unclear how the model works. Is it a cellular automaton? Or just a statistical model? How is the spatial influence of neighboring regions considered? How is function (industrial, residential, ...) considered? How transport?

- There is a severe lack of evaluation of model and in addition very little doubt about model results – these are presented like facts. In line 287 ff. you state: "Though there are no data for validating long-term spatial projections like ours, we thoroughly examined the model's robustness, generalizability, and short- and mid-term performance, and it scored satisfactorily in all tests we ran". I strongly disagree here – given that you have four timesteps of GHSL data it would be an obvious procedure leave one or two out in training and use them to evaluate the model. Also a summary of these previous tests should be reported here.

Moreover you state: "We also compared our projections with conclusions established by existing literature, such as that cities across the world are generally becoming more expansive, i.e. they grow faster in land area than population size [...] Such fidelity between our projections and state-of-the-art understandings of global land change gives confidence in the quality of our results." Frankly, you cannot mean this seriously! Clearly, any model should be able to reproduce the most basic characteristics of urbanization, yet that clearly does not justify great confidence yet. This is like stating the great quality of a model miniature train because it has wheels. Beside this criterion being most basic, it is also evident since the model is trained on data with these characteristics and therefore this is rather trivial.

- The manuscript lacks of perception and consideration of global urban products. You use the GHSL layer – although this is not explicitly stated until Tab. 2 in line 357. This certainly is a valuable resource, but you ignore other recent urban masks like the global urban footprint (Esch et al., 2017), the world settlement footprint (<https://urban-tep.eu/#/>) or ESA CCI land cover - for an overview see (Grekousis et al., 2015). You state that: "We consider our input data the best available for globally consistent spatially-explicit time-series observations of urban land change". I fully agree that GHSL is a rational choice even though not well justified here. However, this dataset is NOT available in decadal timesteps but four discrete epochs: 1975, 1990, 2000, and 2014, which do not match the model timesteps. It is not even mentioned how the data is preprocessed in this respect.

- I find it very doubtful to model urban expansion on the very coarse resolution of 0.125°, since much of urban expansion including different patterns can be assumed but subscale, in fact most cities on the planet are. However, if you think it is suitable (which is not argued or justified) it seems absolutely unclear why you would need 30 m resolution input data to calculate urban fractions for huge grid cells.

- The presentation of Figs (i.e. simple excel plots, partly missing legends (e.g. Fig. 4)) and maps (missing legends (Fig. 3, SI 1, ...), scale bars, north arrows. etc., ArcGIS standard maps) is not only quite poor for a nature paper but lacks basic standards.

[Minor]

- Line 177: suddenly you speak of different urban types (residential, commercial, ...). Where do they come from? Predicted variable is urban fraction without any differentiation, right?

- Line 222: GDP change has not been discussed before and it wasn't even clear that this variable is considered in the model.

- Fig. 4: it is relatively unclear from caption, which data this refers to. Actually, you have to read footnote from Table 1 to assume it is the historic data as well. But which years? Which input data? Which variables were considered in the cluster analysis?

- Line 334 ff.: "In this work, urban land is defined as built-up land (a.k.a. impervious surface, developed land)," -> these are two different definitions, and GHSL is rather impervious than build up!

- Fig. 7: What role plays the national level in the data processing? Is it suitable (i.e. Nations have very different urbanization states within them)?

Mapping Global Urban Land Expansion for 21st Century Using Data Science Based Simulations

The paper provides an empirically based approach for simulating urban expansion at a global scale. Source data sources and methodology are described in a straightforward manner, and the resulting global maps of urban change would certainly be of interest to many parties. However, while the introduction, methods, and results sections are relatively well written, there are key elements missing from the paper.

Chief among these is a lack of any meaningful validation or assessment of model results. I'm completely aware of the difficulties in "validating" long-term projections such as these. However, the current paper pays basic lip service to the idea of model assessment, without actually providing any meaningful results, either quantitatively or qualitatively. The paragraph starting on line 287 is what constitutes the assessment/validation of model performance. In the first line, the authors state they "thoroughly examined the model's robustness, generalizability, and short- and mid-term performance, and it scored satisfactorily in all the test we ran". The authors then fail to provide any of those tests. How can a reader judge model performance if those "satisfactory" test results are not provided? If such tests are available, clearly they need to be part of the paper.

The only real attempt to qualify model performance comes in the next few sentences, where the authors note that the model matches expectations of urban development "growing faster in land area than population size". That's the sum of all provided evidence for model performance, and that's a very low bar to cross. With such a long historical period used to calibrate the model, there clearly should have been ample opportunity to also test model performance. Why didn't the authors test model performance for some period from 1980-2010? In lieu of that most obvious validation opportunity, it is recognized that quantitative model validation is difficult for long-term projections, but the authors could have provided much more evidence as to how their results matched (or didn't match) theoretical patterns of urban growth. There are...many...papers describing theoretical long-term urban growth that could have been referenced, with qualitative comparisons done.

The theoretical basis of urban growth is an overall weakness of the approach. I'm NOT criticizing the approach in general. It seems logical, and an empirically based model of global urban change is certainly a much more straightforward and practical approach than a more involved process-based model. But because it's an empirically based model, the authors sometimes oversell the utility of the approach.

One example where this could have been addressed is in the discussion section. However, the discussion section is woefully inadequate. It's one paragraph of general statements that add little value to the paper. What I'd have liked to have seen in the discussion section was an honest assessment of the strengths and weaknesses of the approach. You modeled urban change for the globe! That's huge! That's a nice accomplishment, so use the discussion section to highlight the practical nature of your approach. But also note the weaknesses of an empirically based approach.

For example...the model is very heavily weighted towards areas that experienced significant urban change from 1980 to 2010. How valid is that for a projection out to 2100? One example is the western United States, particularly the southwestern US. Yes, that area experienced incredibly high growth. Is that sustainable for the next century? As an empirical model based heavily on past urban growth, it

doesn't take into account bottlenecks or other elements that could constrain change in the future. The obvious one is climate change, and the massive impact that could have on coastal regions that currently have incredibly high population densities. Basic water availability is another limiting factor. Water stresses in the southwestern US are already a big issue, and it is curious to see many new big urban centers popping up in these very dry areas in a scenario like SSP5.

These are the kinds of issues that should be discussed in the discussion section. The authors wouldn't need to go into great depth, but they need an honest accounting of the limitations of their approach. An empirical based model just can't deal with non-stationarity of land-change processes, particularly over such a long simulation period. That's fine, but...note that limitation in the discussion.

In short, the authors need to provide a better assessment of model results, whether that's through a formal calibration/validation period for the 1980-2010 historical period, or a much more comprehensive analysis of how their results match (or don't) long-term urban theory. The discussion needs to be completely rewritten, and provide the elements noted above.

Specific comments follow.

Sequential Comments

- Line 24 – “Across scales” is odd on its own. Temporal scales? Spatial scales? Define.
- Line 29 – “Value creation” is an odd term as well. Value of what? It's too generic as is.
- Line 30 – “Societal impacts of environmental stresses” – As phrased it implies it's society impacting environmental parameters, and I think you're trying to state environmental stressor impacts on society (air pollution, disease, etc.). Rephrase.
- Line 33 – When you say “Many” have argued for something, it would be a stronger argument if more than one reference were provided to back up that assertion.
- Paragraph starting on line 33 – It seemed odd when reading to see the three italicized words. Then I saw those were the same key words used at the end of the first sentence of the paragraph. That's a nice structure, but it might help to also italicize the words in the first sentence so a reader makes that link more intuitively?
- Paragraph starting on line 41 – I'm not sure I agree with the assertions at the end of the paragraph. Given the vast number of studies that have used time series analyses to drive forecast models of urban change, it's clearly overstated to say it's “difficult, if not impossible” to incorporate this type of data in forecast models. I get what you're saying...that these data may be available for certain areas, but not globally. Just clarify your argument here and in the next paragraph. Time series data ARE useful and HAVE been widely used for urban modeling, but only for select, local studies (typically). The difficulty is acquiring consistent, global scale time series analyses to drive global scale urban modeling. Make it a little clearer that's the niche of your work here, but don't discount the value of time series analyses for past (local) work. Perhaps note the concepts from those local studies that you're expanding to a global scale application.
- Line 82 – 38 meters...I am curious about Pesaresi et al using coarser-scale MSS and effectively downscaling it to a higher spatial resolution.
- Lines 93-96 – Is there any concern about characterizing all cities in a country with the same “style”? For a large country like the US there are clearly different styles dependent upon location and history of a city.

- Introduction – At some point in the introduction I think it’s important to clarify what you’re classifying as “urban land”. So many remote-sensing approaches are heavily biased towards high-density urban dominated by impervious surface. I’m not familiar with the Pesaresi study, but knowing how Landsat struggles to characterize low-density residential characteristics of many suburban areas, I’m wondering what the “total urban area” of 0.6 million km² in 2000 represents. It’s not until I get down into the methodology that you define what you mean by “urban” (and indeed, it’s imperviousness-centric). Just a phrase is needed in the intro so a reader is aware of this distinction.
- Lines 163-168 – This does surprise me. I don’t doubt you, but given it is a result that may surprise other readers as well, it might help to have a reference or two that backs the point about economically (and historically) well-developed regions like Europe still having the potential to grow this substantially. Perhaps reference #7 does this, since you point to it in a similar context on line 175.
- Lines 177-180 – For the next step of your research! I hesitate to call it a “weakness” of your approach as I appreciate the challenges in what you’re doing, but modeling one “urban” class masks a lot of the story of change, as well as the differing driving forces between residential, industrial, and commercial development.
- Lines 182-184 – I don’t doubt that these factors will continue to drive urban development/patterns even after population stabilization. But the challenge for an empirically based model is that of non-stationarity. These factors will still drive change, but it’s doubtful the same quantitative, empirically derived relationship will remain constant. You do make the same kind of a point later on (paragraph starting on line 212).
- Line 202 – In the introduction and/or methods sections, you need some theoretical basis for this stated trend. Down in the methodology you seem to have used a clustering analysis to identify these land expansion styles. However, that clustering cannot identify a sequential trend from 1, to 2, to 3. There’s been a lot written about theoretical patterns of urban change over time. How does this scheme fit into the existing theoretical literature?
- Line 205 – What is a “more developed phase”? Urban density, or something else?
- Lines 214-217 – Given the result in Table 1, somewhere in the text here for “rapidly urbanizing” countries I’d note the strong negative relationship between urban share of the population, and urban land expansion in these countries.
- Paragraph starting on line 233 – One disadvantage of this paper structure (and I admit it’s a structure I’m not fond of)...without the methodological background, a reader at this point really has no idea how you determine when a country reaches a tipping point, and is categorized into the next urbanization class. It’s thus difficult to put the results in this paragraph into context.
- Line 261 – A personal bias against the word “forecast” for work such as this. Forecast implies prediction. Land-use modeling such as this is scenario based, with the scenarios representing future uncertainties in land use. Forecast is a word I thus tend to avoid. “Project” instead of “forecast”? “Forecast” particularly seems like a poor word choice in a sentence that also includes the phrase “educated guess”. 😊
- Paragraph starting on line 272 – Are the actual patterns different, or is it just a matter of the magnitude of development? From something like Figure 5, the overall patterns of where development are occurring seem the same, it’s just that the magnitude of development is obviously quite different. Are there driving forces in the methodology that actually produce

development in different locations? Where one region may have preferentially (proportionally) more development than another region, depending specifically upon scenario assumptions? Or is it indeed just a matter of magnitude?

- Figure 5 – The spatial patterns here are rather curious to me. It seems odd that the coastal areas (New Jersey, Delaware, Massachusetts, Long Island) experience incredibly high development in a scenario like SSP5. However, other contiguous parts of the big megalopolis that runs from Boston down through Washington DC are evidently immune from that growth. The DC/Baltimore area for example shows curiously low growth in any scenario.
- Paragraph starting on line 287 – This is by far the weakest part of the paper. For this first sentence...what does it even mean that you “thoroughly examined the model’s robustness, generalizability, and short- and mid-term performance, and it scored satisfactorily in all the tests we ran”? What tests? If you have that evidence of the “robustness”, “generalizability”, and “performance”...that evidence absolutely must be provided to the reader. This first sentence is meaningless without the data to back it up. The only evidence provided for model performance is the next few sentences, where you state the model matches expectations of growing “faster in land area than population size”. That’s a low bar to clear, and while it’s good the model matches that expectation, it doesn’t really provide much validation of model performance. The last two sentences of the paragraph are just poor (sorry...love the approach overall, but have to be blunt here). What “state of the art” understanding are you referring to that supposedly highlights the “fidelity” of this work? For the last sentence, you’ve really provide zero evidence of any “performance level”, so thus there’s no evidence of the “power and potential of creative data science applications...”. You need a much more robust assessment of model performance, preferably quantitative in nature, but even a lot more qualitative comparison of results with theoretical patterns of urban growth would be a big improvement.
- Discussion Section – One Paragraph. First off, with the first sentence...How? How do the RESULTS demonstrate the need for understand interactions between urbanization, and changes in society, economies, and the environment? The results are new maps of urban change for the globe. A cool product to be sure! But how does that demonstrate the need for understanding these interactions? Start with something more akin to line 2 of this paragraph, where you note these projections could be used to examine the potential interactions between urban change, and the “interactions” noted above. I could also do without the kind of language in the last 3 sentences of the paragraph. Overall I don’t disagree with some of the sentiment. But the language and phrasing just doesn’t seem appropriate for the journal or topic. More than that, it takes the message away from what you’ve achieved with this work. In sum...the discussion section is disappointing. Instead of using flowery language about making the world a better place, DEMONSTRATE how your work potentially does that. Summarize the weaknesses of existing global urban projections. Point out what your work offers that the older work does not. Mention SPECIFIC examples of how your work could improve understanding of the “interactions” noted in the first part of the paragraph.
- Line 317 – Not fond of the word “extrapolate” here, as that implies a methodology much simpler than what you’ve done. I’d also be a bit careful about touting the ability to model well “over long time horizons” given the limitations in assessing your model performance, as noted above.

- Methods first paragraph and a half – Much of the material here reads like background material, or something that should be in an introduction. Indeed most of those concepts were discussed in the introduction. I'd eliminate everything up through line 321 here (lines 313-321), moving and condensing that material with what you already have in the introduction. Then keep the methods section focused on actual methods, rather than background material.
- Line 324 – Make it clear for a reader what “the two” are. Maybe start the sentence like “To balance the need for long time horizons and higher resolution, we used data...”
- Lines 327-330 – Again a weakness of the structure, where methods come at the end, but as a reader, up until this point, I assumed you were modeling urban land cover at 38m, given the multiple references to (roughly) Landsat scale data and this 38-m resolution dataset. Just needs a phrase or something in the introduction that notes what your model resolution is.
- Line 331-333 – Try to be more specific when describing concepts like this. DID you use an approach that specifically capitalizes on using a numerical response variable? Right now you're phrasing it as a hypothetical improvement, but a reader has no idea if you actually used a “better” methodology, due to use of a numerical response variable.
- Lines 338-341 – You've noted (rightfully so) the difficulties in mapping low-density urban from remote sensing imagery. You're also strongly implying that using Landsat helps overcome that limitation, compared to something like MODIS. It would really help if you had a supporting reference or two that backed this assertion. Landsat too has issues identifying settlements with a high proportion of vegetation.
- Table 2 – Couldn't you update your “distance to existing cities” layer as projections in population tip a city over 300k people?
- Line 360 – In an ideal world you'd have a table or something that shows what the tested metrics were.
- Lines 364-367 – Isn't this rather circular in logic...using national change rates in urban land serves as the best predictor of...changing urban land?
- Lines 370-371 – As noted above, you need to tie this clustering in with published urban theory. Clearly you had some pre-conceived notion of what clusters you were looking for, given that the labels for those 3 clusters had to come from somewhere. It's fine that you used statistical methods for characterizing those clusters, but the whole theoretical underpinning of the categorization (and trend over time as noted previously) should be better explained.
- Line 372 – Why “consistently classified”? The clustering would produce outcomes on a continuum, so it's not as if everything always falls neatly and “consistently” into these 3 clusters.
- Lines 379-382 -- It's still not clear to me what triggers the move of a country's classification to a different category. I assume it's just tied to changes in the SSP projection categories (right column of Table 2? Might be worth a phrase or something here that notes what drives the change in categorization.
- Paragraph starting on line 383 – You have many countries that will move to a new land expansion style in future decades. How do you model country/style combinations, if that style didn't exist during the historical period? Your examples A and B here only cover styles that occurred during the historical period.
- Table 3 – I'm curious how these values were assigned. It's a bit of an enigma for example to see “low” rates of development for the “rapidly urbanizing” style in SSP4, compared to higher rates for the other two styles.

- Line 410 – “Best guess” coefficients? How were these coefficients obtained? Or is “best guess” just a poor choice of phrasing? Given it has such a key influence on the model, some clarity here would be welcome.

Reviewer #1:

[General]

The manuscript “Mapping Global Urban Land Expansion for 21st Century Using Data Science Based Simulations” covers a highly interesting and relevant topic and aims at consistent urban land cover scenarios for the 21st century considering different urbanization pathways and states. It is generally well written, yet the provided materials are insufficient to assess if it is technically sound and I am much more sceptic about the results than the authors.

[Major]

- Relatively little detail is given about the used model and just a reference and various input datasets reported are provided. To me it is relatively unclear how the model works. Is it a cellular automaton? Or just a statistical model? How is the spatial influence of neighboring regions considered? How is function (industrial, residential, ...) considered? How transport?

Edits: We made three major modifications according to this comment: (1) Clarified details of the model/method throughout the paper, especially in “introduction” (lines 92-120) and “methods” (lines 440-652). (2) Clarified what model validations have been conducted (lines 331-341, 548-553), and that most validation results have already been published in a separate, open-access paper (Gao & O’Neill 2019) (lines 118-120, 139-141, 339-341, 598-599, 629), which has been uploaded for the editor and the reviewers’ reference. We apologize for the inconvenience of having to read another paper to evaluate this one, but the size of the work doesn’t fit into one article. Our modeling framework consists of two models (CLUBS that projects national total amount of urban land, and SELECT that models spatial patterns). Both are new development. Documenting all their details and validations is too large a task for a single paper. (3) Further highlighted key differences between our modeling framework and conventional/existing methods throughout the paper, especially in the rewritten “discussion” (lines 357-437), the “introduction” (lines 49-60), and the “method” (lines 595-652).

Additionally, we added clarifications addressing the reviewer’s specific questions here at lines 202-203 (functions of urban land are not explicitly considered), 374-

385 & 599-600 (the model is essentially data-driven but integrates existing theories when possible), 622-632 (influences of road network and spatial autocorrelation are considered through focal metrics).

- There is a severe lack of evaluation of model and in addition very little doubt about model results – these are presented like facts. In line 287 ff. you state: “Though there are no data for validating long-term spatial projections like ours, we thoroughly examined the model’s robustness, generalizability, and short- and mid-term performance, and it scored satisfactorily in all tests we ran”. I strongly disagree here – given that you have four timesteps of GHSL data it would be an obvious procedure leave one or two out in training and use them to evaluate the model. Also a summary of these previous tests should be reported here.

Moreover you state: “We also compared our projections with conclusions established by existing literature, such as that cities across the world are generally becoming more expansive, i.e. they grow faster in land area than population size [...] Such fidelity between our projections and state-of-the-art understandings of global land change gives confidence in the quality of our results.” Frankly, you cannot mean this seriously! Clearly, any model should be able to reproduce the most basic characteristics of urbanization, yet that clearly does not justify great confidence yet. This is like stating the great quality of a model miniature train because it has wheels. Beside this criterion being most basic, it is also evident since the model is trained on data with these characteristics and therefore this is rather trivial.

Edits: For the spatial model SELECT, we have examined, (1) in short-term applications, the model’s residuals and fractions of variations in the response variable explained by the model, (2) in mid-term applications, a comparison between projections made by SELECT and an example existing urban land change model, and (3) to gain confidence in the model’s long-term reliability, we have tested its statistical robustness by examining the model’s reaction to simulated noise, spatial generalizability by swapping models trained for different subnational regions, and temporal generalizability by leaving one decade of data out of model training for independent performance evaluation. The model performed satisfactorily in all these tests and a complete report of these results has been published in Gao and O’Neill 2019. In the modified manuscript, we improved the descriptions about what validations have been conducted, and clarified the reference to the already published results, lines 331-341.

For the national model CLUBS, the R^2 for estimating decadal national urban land change rates (i.e. the model’s response variable) is 0.503, which corresponds to a 0.998 R^2 for estimating the end-of-the-decade national total urban land areas. The high value of the latter R^2 is partially due to the low magnitude of decadal changes in national total urban land areas relative to existing amounts, which is also evidenced by the fact that the R^2 of a “dummy” baseline model that assumes constant national change rates is 0.983 on the same variable. In the modified manuscript, we added these descriptions of the model’s performance at lines 545-553.

About plausibility tests, although fidelity to known patterns at aggregate levels (e.g. the pattern that global cities have been growing faster in land area than population size) should be a basic requirement for projection models, they often cannot be assumed, especially when the time horizon to be projected is much longer than the one in available training data. In the modified manuscript, we clarified why the plausibility tests were used and added more descriptions of our findings, lines 341-356.

- The manuscript lacks of perception and consideration of global urban products. You use the GHSL layer – although this is not explicitly stated until Tab. 2 in line 357. This certainly is a valuable resource, but you ignore other recent urban masks like the global urban footprint (Esch et al., 2017), the world settlement footprint (<https://urban-tep.eu/#!>) or ESA CCI land cover - for an overview see (Grekousis et al., 2015). You state that: “We consider our input data the best available for globally consistent spatially-explicit time-series observations of urban land change”. I fully agree that GHSL is a rational choice even though not well justified here. However, this dataset is NOT available in decadal timesteps but four discrete epochs: 1975, 1990, 2000, and 2014, which do not match the model timesteps. It is not even mentioned how the data is preprocessed in this respect.

Edits: We added a discussion of global urban land mapping products (including the ones highlighted by the reviewer) other than GHSL, at lines 485-490, to explain the data choice. To generate maps for 1980 and 2010, we used temporal linear interpolation on the raw GHSL data, so that the time steps are regular and the time points align with other datasets. In the modified manuscript, we noted this data prepping step at lines 508-510.

- I find it very doubtful to model urban expansion on the very coarse resolution of 0.125° , since much of urban expansion including different patterns can assumed to but subscale, in fact most cities on the planet are. However, if you think it is suitable (which is not argued or justified) it seems absolutely unclear why you would need 30 m resolution input data to calculate urban fractions for huge grid cells.

Edits: We added more explanations about the spatial resolution choice at lines 453-471, 106-107. Briefly, $1/8$ degree is a balance between commonly-used spatial resolutions of global change models (e.g. $0.5-2$ degrees) and urban land change models (e.g. 30-500 m), as well as a balance between different spatial resolutions of available global data. We agree with the reviewer that, if urban land is treated as the commonly-used binary variable (urban vs. non-urban), $1/8$ degree is too coarse. This is why we model the “fraction of urban land within each grid” to maintain spatial precision, so that sub-grid change/growth can be captured. The 38 m base data are helpful to give the fraction estimates the most precision currently possible.

- The presentation of Figs (i.e. simple excel plots, partly missing legends (e.g. Fig. 4)) and maps (missing legends (Fig. 3, SI 1, ...), scale bars, north arrows. etc., ArcGIS standard maps) is not only quite poor for a nature paper but lacks basic standards.

Edits: We made a few modifications according to the reviewer's suggestions: (1) Modified the legends for Figure 4, and SI Figure 2. (2) Moved the legend of Figure 3 to a more prominent position (from the right side to the bottom of the graphics). (3) Added north arrow and scale bar for Figure 5 (the other maps are of continental to global scales and don't seem to need indications of such information).

[Minor]

- Line 177: suddenly you speak of different urban types (residential, commercial, ...). Where do they come from? Predicted variable is urban fraction without any differentiation, right?

Edits: Yes, the response variable is the fraction of urban land. This sentence mentions the functional urban land types to assist the discussion of two potential reasons (starting line 198) why urban land would continue expanding in developed countries after total population size stabilizes. We clarified these points at lines 202-203.

- Line 222: GDP change has not been discussed before and it wasn't even clear that this variable is considered in the model.

Edits: Added a discussion of drivers considered by the models at the beginning of the paragraph, lines 246-250.

- Fig. 4: it is relatively unclear from caption, which data this refers to. Actually, you have to read footnote from Table 1 to assume it is the historic data as well. But which years? Which input data? Which variables were considered in the cluster analysis?

Edits: Changed the caption to "Three styles / stages of urban land expansion: scatter plot of decadal observations of global countries, showing data from three decades (1980-2010). (The two axis variables are shown for visual clarity. The actual classification step in the modeling framework uses more variables. More information in methods.) (Source data are provided as a Source Data file.)", lines 241-245.

- Line 334 ff.: "In this work, urban land is defined as built-up land (a.k.a. impervious surface, developed land)," -> these are two different definitions, and GHSL is rather impervious than build up!

Edits: Changed (lines 475-478) to "In this work, urban land is defined as the Earth's surface that is covered primarily by manmade materials, such as cement, asphalt, steel, glass, etc. This land cover type is referred to by different communities using different terms (with nuances), such as built-up land, impervious surface, and developed land."

- Fig. 7: What role plays the national level in the data processing? Is it suitable (i.e. Nations have very different urbanization states within them)?

Edits: Clarified labels in and captions of the figure, lines 590-593. Added clarifications about how subnational variations are modeled in the “introduction” section, lines 108-118.

Reviewer #2:

The paper provides an empirically based approach for simulating urban expansion at a global scale. Source data sources and methodology are described in a straightforward manner, and the resulting global maps of urban change would certainly be of interest to many parties. However, while the introduction, methods, and results sections are relatively well written, there are key elements missing from the paper.

Chief among these is a lack of any meaningful validation or assessment of model results. I’m completely aware of the difficulties in “validating” long-term projections such as these. However, the current paper pays basic lip service to the idea of model assessment, without actually providing any meaningful results, either quantitatively or qualitatively. The paragraph starting on line 287 is what constitutes the assessment/validation of model performance. In the first line, the authors state they “thoroughly examined the model’s robustness, generalizability, and short- and mid-term performance, and it scored satisfactorily in all the test we ran”. The authors then fail to provide any of those tests. How can a reader judge model performance if those “satisfactory” test results are not provided? If such tests are available, clearly they need to be part of the paper.

The only real attempt to qualify model performance comes in the next few sentences, where the authors note that the model matches expectations of urban development “growing faster in land area than population size”. That’s the sum of all provided evidence for model performance, and that’s a very low bar to cross. With such a long historical period used to calibrate the model, there clearly should have been ample opportunity to also test model performance. Why didn’t the authors test model performance for some period from 1980-2010? In lieu of that most obvious validation opportunity, it is recognized that quantitative model validation is difficult for long-term projections, but the authors could have provided much more evidence as to how their results matched (or didn’t match) theoretical patterns of urban growth. There are...many...papers describing theoretical long-term urban growth that could have been referenced, with qualitative comparisons done.

Edits: We clarified details of the model/method according to reviewers’ comments throughout the paper, especially in “introduction” (lines 92-120) and “methods” (lines 440-652).

We clarified what model validations have been conducted (more explanations below) (lines 331-341, 548-553), and that most validation results have already been published in a separate, open-access paper (Gao & O’Neill 2019) (lines 118-

120, 139-141, 339-341, 598-599, 629), which has been **uploaded** for the editor and the reviewers' reference. We apologize for the inconvenience of having to read another paper to evaluate this one, but the size of the work doesn't fit into one article. Our modeling framework consists of two models (CLUBS that projects national total amount of urban land, and SELECT that models spatial patterns). Both are new development. Documenting all their details and validations is too large a task for a single paper.

For the spatial model SELECT, we have examined, (1) in short-term applications, the model's residuals and fractions of variations in the response variable explained by the model, (2) in mid-term applications, a comparison between projections made by SELECT and an example existing urban land change model, and (3) to gain confidence in the model's long-term reliability, we have tested its statistical robustness by examining the model's reaction to simulated noise, spatial generalizability by swapping models trained for different subnational regions, and temporal generalizability by leaving one decade of data out of model training for independent performance evaluation. The model performed satisfactorily in all these tests and a complete report of these results has been published in Gao and O'Neill 2019. In the modified manuscript, we improved the descriptions about what validations have been conducted, and clarified the reference to the already published results, lines 331-341.

For the national model CLUBS, the R^2 for estimating decadal national urban land change rates (i.e. the model's response variable) is 0.503, which corresponds to a 0.998 R^2 for estimating the end-of-the-decade national total urban land areas. The high value of the latter R^2 is partially due to the low magnitude of decadal changes in national total urban land areas relative to existing amounts, which is also evidenced by the fact that the R^2 of a "dummy" baseline model that assumes constant national change rates is 0.983 on the same variable. In the modified manuscript, we added these descriptions of the model's performance at lines 545-553.

About plausibility tests, although fidelity to known patterns at aggregate levels (e.g. the pattern that global cities have been growing faster in land area than population size) should be a basic requirement for projection models, they often cannot be assumed, especially when the time horizon to be projected is much longer than the one in available training data. In the modified manuscript, we clarified why the plausibility tests were used and added more descriptions of our findings, lines 341-356.

Additionally, we further highlighted key differences between our modeling framework and conventional/existing methods throughout the paper, and added discussions about how this work connects with existing urban theories wherever relevant, especially in the rewritten "discussion" section (lines 357-437), and the "introduction" (lines 49-60).

The theoretical basis of urban growth is an overall weakness of the approach. I'm NOT criticizing the approach in general. It seems logical, and an empirically based model of global urban change is certainly a much more straightforward and practical approach than a more involved process-based model. But because it's an empirically based model, the authors sometimes oversell the utility of the approach.

One example where this could have been addressed is in the discussion section. However, the discussion section is woefully inadequate. It's one paragraph of general statements that add little value to the paper. What I'd have liked to have seen in the discussion section was an honest assessment of the strengths and weaknesses of the approach. You modeled urban change for the globe! That's huge! That's a nice accomplishment, so use the discussion section to highlight the practical nature of your approach. But also note the weaknesses of an empirically based approach.

For example...the model is very heavily weighted towards areas that experienced significant urban change from 1980 to 2010. How valid is that for a projection out to 2100? One example is the western United States, particularly the southwestern US. Yes, that area experienced incredibly high growth. Is that sustainable for the next century? As an empirical model based heavily on past urban growth, it doesn't take into account bottlenecks or other elements that could constrain change in the future. The obvious one is climate change, and the massive impact that could have on coastal regions that currently have incredibly high population densities. Basic water availability is another limiting factor. Water stresses in the southwestern us are already a big issue, and it is curious to see many new big urban centers popping up in these very dry areas in a scenario like SSP5.

These are the kinds of issues that should be discussed in the discussion section. The authors wouldn't need to go into great depth, but they need an honest accounting of the limitations of their approach. An empirical based model just can't deal with non-stationarity of land-change processes, particularly over such a long simulation period. That's fine, but...note that limitation in the discussion.

In short, the authors need to provide a better assessment of model results, whether that's through a formal calibration/validation period for the 1980-2010 historical period, or a much more comprehensive analysis of how their results match (or don't) long-term urban theory. The discussion needs to be completely rewritten, and provide the elements noted above.

Edits: We completely rewrote the “discussion” section according to the reviewer’s comments (lines 357-437). We agree with the reviewer that given the complexities of modeling global, long-term urban land change, any approach to such modeling efforts will have both strengths and limitations. To maximally take advantage of all information available, our method integrates existing theories and new data. On balance, the model is data-driven, because the newly available time series of global spatial urban land we use here offer perspectives that were not possible before, and we therefore allowed observational evidence from the

data analyses to extend and update existing theories. However, we view the perspectives offered by existing theories and new data as a gradient of concepts rather than a black-and-white distinction. In the rewritten “discussion”, we discussed in depth two examples of how existing theories and new data are combined in our method (while the integrative principal was applied throughout our model design). We also added discussions about how our method approaches temporal non-stationarity and alternative scenarios in the new “discussion” section.

Specific Comments

- Line 24 – “Across scales” is odd on its own. Temporal scales? Spatial scales? Define.

Edits: Changed to “across spatial and temporal scales”.

- Line 29 – “Value creation” is an odd term as well. Value of what? It’s too generic as is.

Edits: Changed to “economic value creation”.

- Line 30 – “Societal impacts of environmental stresses” – As phrased it implies it’s society impacting environmental parameters, and I think you’re trying to state environmental stressor impacts on society (air pollution, disease, etc.). Rephrase.

Edits: Changed to “The spatial distribution of urban land also shapes the societal impacts of environmental stresses [...]”.

- Line 33 – When you say “Many” have argued for something, it would be a stronger argument if more than one reference were provided to back up that assertion.

Edits: Added more reference.

- Paragraph starting on line 33 – It seemed odd when reading to see the three italicized words. Then I saw those were the same key words used at the end of the first sentence of the paragraph. That’s a nice structure, but it might help to also italicize the words in the first sentence so a reader makes that link more intuitively?

Edits: Italicized the words as suggested.

- Paragraph starting on line 41 – I’m not sure I agree with the assertions at the end of the paragraph. Given the vast number of studies that have used time series analyses to drive forecast models of urban change, it’s clearly overstated to say it’s “difficult, if not impossible” to incorporate this type of data in forecast models. I get what you’re saying...that these data may be available for certain areas, but not globally. Just clarify your argument here and in the next paragraph. Time series data ARE useful and HAVE been widely used for urban modeling, but only for select, local studies (typically). The difficulty is acquiring consistent, global scale time series analyses to drive global scale urban modeling. Make it a little clearer that’s the niche of your work here, but don’t discount the value of time series analyses for past (local) work. Perhaps note the concepts from those local studies that you’re expanding to a global scale application.

Edits: Clarified the text to highlight that the focus of our discussion is large-scale (global, long-term) modeling, lines 49-53.

- Line 82 – 38 meters...I am curious about Pesaresi et al using coarser-scale MSS and effectively downscaling it to a higher spatial resolution.

- Lines 93-96 – Is there any concern about characterizing all cities in a country with the same “style”? For a large country like the US there are clearly different styles dependent upon location and history of a city.

Edits: Subnational variations are accounted for in our modeling framework by the spatial model component (SELECT), which divides the world into 375 subnational regions and models the regions separately. The continental U.S. is actually modeled as 28 separate regions. We modified the text to improve clarity, lines 94-97, 108-118.

- Introduction – At some point in the introduction I think it's important to clarify what you're classifying as "urban land". So many remote-sensing approaches are heavily biased towards high-density urban dominated by impervious surface. I'm not familiar with the Pesaresi study, but knowing how Landsat struggles to characterize low-density residential characteristics of many suburban areas, I'm wondering what the "total urban area" of 0.6 million km² in 2000 represents. It's not until I get down into the methodology that you define what you mean by "urban" (and indeed, it's imperviousness-centric). Just a phrase is needed in the intro so a reader is aware of this distinction.

Edits: Added a brief definition of urban land to the introduction "(defined as built-up land)", line 88.

- Lines 163-168 – This does surprise me. I don't doubt you, but given it is a result that may surprise other readers as well, it might help to have a reference or two that backs the point about economically (and historically) well-developed regions like Europe still having the potential to grow this substantially. Perhaps reference #7 does this, since you point to it in a similar context on line 175.

Edits: We haven't seen existing literature reporting this pattern (i.e. economically developed regions have not stopped building new urban land). However, our own analyses of historical data support the pattern very well. We added an example from our analysis results at lines 185-191 as additional support. "According to our own analysis of time-series observational data, during the decade of 2000-2010, more than 15 thousand km² of new urban land was built in Europe (excluding Russia), and 17 thousand km² in Africa. These similar amounts of urban land development occurred despite the fact that in 2000, Europe already had more than 144 thousand km² urban land (with about 0.6 billion people), while Africa had only about 46 thousand km² urban land (with about 0.7 billion people)."

- Lines 177-180 – For the next step of your research! I hesitate to call it a "weakness" of your approach as I appreciate the challenges in what you're doing, but modeling one "urban" class masks a lot of the story of change, as well as the differing driving forces between residential, industrial, and commercial development.

Edits: Added acknowledgement of this point at lines 202-203.

- Lines 182-184 – I don't doubt that these factors will continue to drive urban development/patterns even after population stabilization. But the challenge for an empirically based model is that of non-stationarity. These factors will still drive change, but it's doubtful the same quantitative, empirically derived relationship will remain constant. You do make the same kind of a point later on (paragraph starting on line 212).

Edits: As mentioned earlier, we have added an explicit discussion about temporal non-stationarity in the rewritten "discussion", lines 408-420.

- Line 202 – In the introduction and/or methods sections, you need some theoretical basis for this stated trend. Down in the methodology you seem to have used a clustering analysis to identify these land expansion styles. However, that clustering cannot identify a sequential trend from 1, to 2, to 3. There's been a lot written about theoretical patterns of urban change over time. How does this scheme fit into the existing theoretical literature?

Edits: In the rewritten “discussion” section, we used the identification and the modeling of different urbanization styles (and transitions among them) in the national-level model (CLUBS) as an example to demonstrate how we combined existing theories with insights from new data, lines 386-395. It's true that the clustering analysis itself does not explicitly put the three urbanization styles in a sequence, but after we label global countries in historical times using the clustering algorithm, we can see that when countries changed styles, the changes are directional (from “rapidly urbanizing” to “steadily urbanizing” to “urbanized”). We now added SI table 2 to show these historical trends.

Additionally, in our model, as a country's urbanization matures, its socioeconomic conditions change, and these changes naturally lead to the labeling of a different urbanization style. We see it as a strength that such transitions are organically modeled in response to how a country's conditions evolve, rather than having the analyst explicitly specify transition rules, because this way more fluidity is allowed for model estimations to respond to different trends in drivers under different scenarios (see for example, the trajectories of China's urbanization styles under different scenarios shown in SI table 3).

- Line 205 – What is a “more developed phase”? Urban density, or something else?

Edits: Changed to “more stabilized urbanization phases with lower yet steady urban change rates”, line 236.

- Lines 214-217 – Given the result in Table 1, somewhere in the text here for “rapidly urbanizing” countries I'd note the strong negative relationship between urban share of the population, and urban land expansion in these countries.

Edits: Added acknowledgement at lines 252-254.

- Paragraph starting on line 233 – One disadvantage of this paper structure (and I admit it's a structure I'm not fond of)...without the methodological background, a reader at this point really has no idea how you determine when a country reaches a tipping point, and is categorized into the next urbanization class. It's thus difficult to put the results in this paragraph into context.

Edits: Edited the methodological overview in “introduction” to provide context for how countries change from one urbanization style to another, lines 99-103.

- Line 261 – A personal bias against the word “forecast” for work such as this. Forecast implies prediction. Land-use modeling such as this is scenario based, with the scenarios representing future uncertainties in land use. Forecast is a word I thus tend to avoid.

“Project” instead of “forecast”? “Forecast” particularly seems like a poor word choice in a sentence that also includes the phrase “educated guess”. ☺

Edits: Changed to “project”.

- Paragraph starting on line 272 – Are the actual patterns different, or is it just a matter of the magnitude of development? From something like Figure 5, the overall patterns of where development are occurring seem the same, it's just that the magnitude of

development is obviously quite different. Are there driving forces in the methodology that actually produce development in different locations? Where one region may have preferentially (proportionally) more development than another region, depending specifically upon scenario assumptions? Or is it indeed just a matter of magnitude?

Edits: The spatial patterns are actually different. We added clarifying texts at lines 316-321, to point out some distinct differences between the spatial patterns projected for different scenarios and briefly discussed how they manifested.

- Figure 5 – The spatial patterns here are rather curious to me. It seems odd that the coastal areas (New Jersey, Delaware, Massachusetts, Long Island) experience incredibly high development in a scenario like SSP5. However, other contiguous parts of the big megalopolis that runs from Boston down through Washington DC are evidently immune from that growth. The DC/Baltimore area for example shows curiously low growth in any scenario.

Response: It does appear the DC/Baltimore area experienced less drastic expansion in SSP 5 than other city centers in the region, but the absolute amount of new development in the area is still substantial, when contrasting SSP5 2100 with SSP5 2030 and SSP1 2100. The most likely cause is that coastal area in this region experienced more new land development during the model training period, and the effects accumulated over time in projections and led to a sizeable difference in an extreme scenario like SSP5 in the end of the century. We double checked whether the model mistakenly treated the DC/Baltimore area differently from other city centers in the region, and that was not the case.

- Paragraph starting on line 287 – This is by far the weakest part of the paper. For this first sentence...what does it even mean that you “thoroughly examined the model’s robustness, generalizability, and short- and mid-term performance, and it scored satisfactorily in all the tests we ran”? What tests? If you have that evidence of the “robustness”, “generalizability”, and “performance”...that evidence absolutely must be provided to the reader. This first sentence is meaningless without the data to back it up. The only evidence provided for model performance is the next few sentences, where you state the model matches expectations of growing “faster in land area than population size”. That’s a low bar to clear, and while it’s good the model matches that expectation, it doesn’t really provide much validation of model performance. The last two sentences of the paragraph are just poor (sorry...love the approach overall, but have to be blunt here). What “state of the art” understanding are you referring to that supposedly highlights the “fidelity” of this work? For the last sentence, you’ve really provide zero evidence of any “performance level”, so thus there’s no evidence of the “power and potential of creative data science applications...”. You need a much more robust assessment of model performance, preferably quantitative in nature, but even a lot more qualitative comparison of results with theoretical patterns of urban growth would be a big improvement.

Edits: As mentioned earlier, we have improved the presentation of our model validation results according to the reviewer’s comments.

- Discussion Section – One Paragraph. First off, with the first sentence...How? How do the RESULTS demonstrate the need for understand interactions between urbanization, and changes in society, economies, and the environment? The results are new maps of urban change for the globe. A cool product to be sure! But how does that demonstrate

the need for understanding these interactions? Start with something more akin to line 2 of this paragraph, where you note these projections could be used to examine the potential interactions between urban change, and the “interactions” noted above. I could also do without the kind of language in the last 3 sentences of the paragraph. Overall I don’t disagree with some of the sentiment. But the language and phrasing just doesn’t seem appropriate for the journal or topic. More than that, it takes the message away from what you’ve achieved with this work. In sum...the discussion section is disappointing. Instead of using flowery language about making the world a better place, DEMONSTRATE how your work potentially does that. Summarize the weaknesses of existing global urban projections. Point out what your work offers that the older work does not. Mention SPECIFIC examples of how your work could improve understanding of the “interactions” noted in the first part of the paragraph.

Edits: The “discussion” section has been rewritten according to the reviewer’s comments.

- Line 317 – Not fond of the word “extrapolate” here, as that implies a methodology much simpler than what you’ve done. I’d also be a bit careful about touting the ability to model well “over long time horizons” given the limitations in assessing your model performance, as noted above.

Edits: Changed to “function well over long time horizons”.

- Methods first paragraph and a half – Much of the material here reads like background material, or something that should be in an introduction. Indeed most of those concepts were discussed in the introduction. I’d eliminate everything up through line 321 here (lines 313-321), moving and condensing that material with what you already have in the introduction. Then keep the methods section focused on actual methods, rather than background material.

Edits: Rewrote and condensed the texts as suggested, starting line 440.

- Line 324 – Make it clear for a reader what “the two” are. Maybe start the sentence like “To balance the need for long time horizons and higher resolution, we used data...”

Edits: Changed according to the reviewer’s comment.

- Lines 327-330 – Again a weakness of the structure, where methods come at the end, but as a reader, up until this point, I assumed you were modeling urban land cover at 38m, given the multiple references to (roughly) Landsat scale data and this 38-m resolution dataset. Just needs a phrase or something in the introduction that notes what your model resolution is.

Edits: Added information about the model’s spatial resolution in “introduction”, lines 106-107.

- Line 331-333 – Try to be more specific when describing concepts like this. DID you use an approach that specifically capitalizes on using a numerical response variable? Right now you’re phrasing it as a hypothetical improvement, but a reader has no idea if you actually used a “better” methodology, due to use of a numerical response variable.

Edits: Using a numerical response variable is a unique difference between our model and conventional models, but the practice by itself is not necessarily an improvement. We modified the text to clarify these ideas, lines 465-475.

- Lines 338-341 – You’ve noted (rightfully so) the difficulties in mapping low-density urban from remote sensing imagery. You’re also strongly implying that using Landsat helps overcome that limitation, compared to something like MODIS. It would really help

if you had a supporting reference or two that backed this assertion. Landsat too has issues identifying settlements with a high proportion of vegetation.

Edits: We think Landsat improves (but not completely overcomes) the limitation, in comparison to MODIS. We clarified the texts at lines 483-486 to better communicate this idea.

- Table 2 – Couldn't you update your "distance to existing cities" layer as projections in population tip a city over 300k people?

Response: Although this might be doable in principle, the actual execution will require a separate project of its own, because the spatial population data are gridded and no globally consistent datasets are available for local administrative boundaries.

- Line 360 – In an ideal world you'd have a table or something that shows what the tested metrics were.

Response: We gave examples of these metrics in the sentence now in lines 514-517. For urban land, we listed 6 variables for 6 possible time epochs (1980-1990, 1990-2000, 2000-2010, 1980-2000, 1990-2010, 1980-2010) – that's 6*6 = 36 variables, and similar measurements for GDP, population size, and urban pop share are imaginable. Since we have already illustrated the types of variables by example, we did not provide what would be a very long table.

- Lines 364-367 – Isn't this rather circular in logic...using national change rates in urban land serves as the best predictor of...changing urban land?

Edits: Now at lines 517-522. The text has been changed to "We found that the most robust statistical relationships across space and time occur among national change rates of urban land, demographic change, and economic growth (i.e. how fast each variable changes at national level) measured over 10-year intervals, rather than the commonly-used per capita urban land and per capita GDP."

- Lines 370-371 – As noted above, you need to tie this clustering in with published urban theory. Clearly you had some pre-conceived notion of what clusters you were looking for, given that the labels for those 3 clusters had to come from somewhere. It's fine that you used statistical methods for characterizing those clusters, but the whole theoretical underpinning of the categorization (and trend over time as noted previously) should be better explained.

Edits: Clarifications about how the three clusters connect with existing theories have been provided in the rewritten "discussion" section, lines 386-395.

- Line 372 – Why "consistently classified"? The clustering would produce outcomes on a continuum, so it's not as if everything always falls neatly and "consistently" into these 3 clusters.

Edits: Removed "consistently".

- Lines 379-382 -- It's still not clear to me what triggers the move of a country's classification to a different category. I assume it's just tied to changes in the SSP projection categories (right column of Table 2? Might be worth a phrase or something here that notes what drives the change in categorization.

Edits: Now at lines 536-538. Clarified that the classification of a country's urbanization style changes "in response to how the country's urbanization maturity evolves."

- Paragraph starting on line 383 – You have many countries that will move to a new land

expansion style in future decades. How do you model country/style combinations, if that style didn't exist during the historical period? Your examples A and B here only cover styles that occurred during the historical period.

Edits: The rewritten discussion section now discusses temporal non-stationarity, lines 408-420.

- Table 3 – I'm curious how these values were assigned. It's a bit of an enigma for example to see "low" rates of development for the "rapidly urbanizing" style in SSP4, compared to higher rates for the other two styles.

Edits: The values were assigned according to our interpretations of the SSP narratives. We added clarifications about the method at lines 128-131, 555-568. Especially, we highlighted the case for SSP 4 which is less intuitive in comparison to other SSPs, lines 562-568.

- Line 410 – "Best guess" coefficients? How were these coefficients obtained? Or is "best guess" just a poor choice of phrasing? Given it has such a key influence on the model, some clarity here would be welcome.

Edits: Changed to "estimated coefficients", line 579.

Reviewers' comments:

Reviewer #1 (Remarks to the Author):

I see some efforts to improve the manuscript and the shortcomings were partly resolved during the revisions. Yet I am still not fully convinced by the evaluation of the model. In the Gao and O'Neill (2019) you state that "The challenge for evaluating temporal generalizability is that only three decades of observational data are available. To utilize as much temporal information as possible in the model, the first two decades were used to generate explanatory variables for estimating change over the third decade, and then no data is left for independent temporal validation". Yet this is not the case, since GHSL is available from 1975 to 2014, which is close enough to 4 decades. Thus, I still think the model should be evaluated based on previous land cover change. Moreover, I still think the quality of figures, in particular the cartography, needs to be improved.

Reviewer #1:

I see some efforts to improve the manuscript and the shortcomings were partly resolved during the revisions. Yet I am still not fully convinced by the evaluation of the model. In the Gao and O'Neill (2019) you state that "The challenge for evaluating temporal generalizability is that only three decades of observational data are available. To utilize as much temporal information as possible in the model, the first two decades were used to generate explanatory variables for estimating change over the third decade, and then no data is left for independent temporal validation". Yet this is not the case, since GHSL is available from 1975 to 2014, which is close enough to 4 decades. Thus, I still think the model should be evaluated based on previous land cover change.

Edits & Response: Our evaluation of the model's temporal generalizability was indeed based on "previous land cover change", but using 3 decades rather than 4 decades of data. We clarified the involvement of "historical" data at line 327.

About GHSL's time span, the dataset directly measures urban land change for 3 time epochs: 1975-1990 (15 yrs), 1990-2000 (10 yrs), 2000-2014 (14 yrs), with 4 time points of observations. For training models that produce temporally-evolving future projections, regular time epochs (e.g. decadal) are necessary. We hence linearly scaled the start and the end points of the time series to be 1980 and 2010, resulting in 3 decades (1980-2010) of model input. This approach maintains all temporal change information (e.g. change rate) that the raw dataset captured, with minimal pre-processing. In contrast, treating 1975-2014 as 4 decadal epochs requires 5 time points of data (which is one more than what the dataset offers – 4 time points of observations). This means one time point must be analytically generated without observational foundation, and therefore the approach requires overall more temporal interpolation (than the approach we used). Such pre-processing does not increase the information volume of the raw data, but introduces more uncertainties. Furthermore, treating the urban land data as 4 decades means the time series have to start mid-decade which misaligns with the time series of many other variables in the model which start at the beginning of decades. These considerations are recognized at lines 469-471.

We reiterate that our model evaluation was based on previous land cover change over 3 decades. Detailed steps of the evaluation are described in Gao & O'Neill

2019 (section 2.4.2), and results of the evaluation are presented at the end of section 3.1 also in that paper. Briefly, the model showed reasonable level of temporal generalizability.

Moreover, I still think the quality of figures, in particular the cartography, needs to be improved.

Edits & Response: We followed all specific suggestions about figures from the last round of review, and are glad to see the reviewer now finds the non-geographic figures satisfactory. The manuscript contains three map-base figures (figures 2, 3, and 5) (the two SI figures are extensions of these three, and hence follow the same design): Figure 5 was modified according to suggestions last round. We changed figure 3 this round to use a multi-panel presentation (now at lines 219-223), considering the previous table-like presentation included unnecessary graph borders. Figure 2, using a classic choropleth design, was intended to be familiar for most readers and easy to interpret.

About the color scheme used in the gridded urban land maps (figures 3 & 5), we explored many options when initially developing the manuscript. Because urban lands usually cluster spatially in small areas relative to the Earth's entire land surface, when maps of regional or global coverages are fitted to typical print or screen sizes, their visuals often give a binary impression (urban or not). In this work, we model the fraction of urban land within each grid cell, and the resulting projections show the spatial gradient of urban/rural transition. Hence, when choosing a color scheme for these maps, we focused on what can better illustrate the spatial gradient of "urban-ness". In the attached "map legend exploration" document (2 pages including 16 maps), we show some color schemes we tried: The one we chose in the end is shown at the top of the page, the next row shows some commonly-used color schemes, and the following rows show some less conventional options. We found that, for communicating the spatial variations of medium-to-low urban land development densities (see eastern U.S. in the graphs as an example), (a) the commonly-used color schemes work better than the unusual options, and (b) our final choice employing a continuum of three hues (yellow-green-navy) works better than color schemes using only one or two hues (e.g. the color schemes shown in the first row in the document: blue, red, yellow-brown).

REVIEWERS' COMMENTS:

Reviewer #1 (Remarks to the Author):

I still disagree on the validation part. Essentially your argument is, that if you used the 4 available decades, the years would be not be divisible by ten (i.e. 1975, 1985, 1995, 2005 & 2015 = 2014 instead of 80/90/00/10) and you would have to reprocess/interpolate the other inputs as well. In addition you state, that you would need to interpolate the land-cover data, but this you currently do as well.

" In contrast, treating 1975-2014 as 4 decadal epochs requires 5 time points of data (which is one more than what the dataset offers – 4 time points of observations). This means one time point must be analytically generated without observational foundation, and therefore the approach requires overall more temporal interpolation (than the approach we used)."

I think none of this is an adequate justification for not evaluating the model based on previous land-cover change.

Reviewer #2 (Remarks to the Author):

Editor - Note comments to the editor

Reviewer #1:

I still disagree on the validation part. Essentially your argument is, that if you used the 4 available decades, the years would be not be divisible by ten (i.e. 1975, 1985, 1995, 2005 & 2015 = 2014 instead of 80/90/00/10) and you would have to reprocess/interpolate the other inputs as well. In addition you state, that you would need to interpolate the land-cover data, but this you currently do as well.

" In contrast, treating 1975-2014 as 4 decadal epochs requires 5 time points of data(which is one more than what the dataset offers –4time points of observations). This means one time point must be analytically generated without observational foundation,and therefore the approach requires overallmore temporal interpolation (than the approach we used)."

I think none of this is a adequate justification for not evaluating the model based on previous land-cover change.

Response: It is incorrect to say the model is not evaluated based on previous land-cover change. As detailed in our responses to previous rounds of reviews, the model was evaluated using GHSL historical land cover data for the period 1975-2014. The current manuscript describes this evaluation at lines 361-366.

The reviewer also does not appear to fully appreciate why we took the specific approach to use the historical data. Our choice to use the four time points in the GHSL data to characterize three periods of time, rather than four, is based on what we believe to be a very strong rationale: minimizing the amount of modification to the data before using it in model evaluation. The table below (on page 2 of this document) summarizes the interpolation required by using the data to represent either 3 or 4 decades of change.

	Native Time Points	when used as 3 decades: Number of Time Points to Interpolate (out of 1980, 1990, 2000, 2010)	when used as 4 decades: Number of Time Points to Interpolate (out of 1975, 1985, 1995, 2005, 2015)
GHSL (land cover)	1975, 1990, 2000, 2014	2 time points (1980, 2010)	3 time points (1985, 1995, 2005)
Other Variables in the Model	1980, 1990, 2000, 2010	none	all five time points (some datasets do not have pre-1975 and/or post-2015 time points to help interpolate for 1975 and 2015)

Clearly, the total amount of temporal interpolation required by treating existing data as 4 decades is substantially more than treating them as 3 decades. In data-driven analyses, less and simpler preprocessing is generally preferred in order to reduce analytically-added uncertainties. We therefore use the data as three rather than four decades.

Moreover, as described in Gao & O'Neill 2019 (i.e. Reference 21) (sections 2.4.2 and 3.1), the 3-decade-based model evaluation results clearly support the conclusion that the model shows a reasonable level of temporal generalizability.